# TIME-VARYING PROPENSITY SCORE TO BRIDGE THE GAP BETWEEN THE PAST AND PRESENT

**Rasool Fakoor**[1], **Jonas Mueller**[2], **Zachary C. Lipton**[3], **Pratik Chaudhari**[1], **Alexander J. Smola**[5]

[1] Amazon Web Services, [2] Cleanlab, [3] Carnegie Mellon University,[5] Boson AI

## ABSTRACT

Real-world deployment of machine learning models is challenging because data evolves over time. While no model can work when data evolves in an arbitrary fashion, if there is some pattern to these changes, we might be able to design methods to address it. This paper addresses situations when data evolves gradually. We introduce a time-varying propensity score that can detect gradual shifts in the distribution of data which allows us to selectively sample past data to update the model—not just similar data from the past like that of a standard propensity score but also data that evolved in a similar fashion in the past. The time-varying propensity score is quite general: we demonstrate different ways of implementing it and evaluate it on a variety of problems ranging from supervised learning (e.g., image classification problems) where data undergoes a sequence of gradual shifts, to reinforcement learning tasks (e.g., robotic manipulation and continuous control) where data shifts as the policy or the task changes.

## 1 INTRODUCTION

Machine learning models are not expected to perform well when the test data is from a different distribution than the training data. There are many techniques to mitigate the consequent deterioration in performance (Heckman, 1979; Shimodaira, 2000; Huang et al., 2006; Bickel et al., 2007; Sugiyama et al., 2007b; 2008; Gretton et al., 2008). These techniques use the propensity score between the train and test data distributions to reweigh the data—and they work well (Agarwal et al., 2011; Wen et al., 2014; Reddi et al., 2015b; Fakoor et al., 2020c;a; Tibshirani et al., 2019) when dealing with a *single* training and test dataset. But when machine learning models are deployed for real-world problems, they do not just undergo one distribution shift[1] (from train to test), but instead suffer many successive distribution shifts (Lu et al., 2019) e.g., search queries to an online retailer or a movie recommendation service evolve as fashion and tastes of the population evolve, etc. Even for problems in healthcare, there are many situations where data drifts gradually, e.g., the different parts of the brain atrophy slowly as the brain ages for both healthy subjects and those with dementia; tracking these changes and distinguishing between them is very useful for staging the disease and deciding treatments (Saito et al., 2022; Wang et al., 2023). Viruses can mutate and drift over time, which can make them resistant to existing treatments (Russell, 2016; Callaway, 2020; Harvey et al., 2021).

To build a method that can account for changes in data, it is important to have some understanding of how data evolves. If data evolves in an arbitrary fashion, there is little that we can do to ensure that the model can predict accurately at the next time instant in general. Settings in which the data distribution drifts gradually over time (rather than experiencing abrupt changes) are particularly ubiquitous (Shalev-Shwartz, 2012; Lu et al., 2019). Assuming observations are not statistically dependent over time (as in time-series), how to best train models in such streaming/online learning settings remains a key question (Lu et al., 2019). Some basic options include: fitting/updating the model to only the recent data (which is statistically suboptimal if the distribution has not significantly shifted) or fitting the model to all previous observations (which leads to biased estimates if shift has occurred). Here we consider a less crude approach in which past data are weighted during training with continuous-valued weights that vary over time. Our proposed estimator of these weights generalizes standard two-sample

---

[1]In this paper, we will not distinguish between covariate, label, and concept shifts; they will all be referred to as distribution shifts and the difference will be clear from context.

propensity scores, allowing the training process to selectively emphasize past data collected at time $t$ based on the distributional similarity between the present and time $t$.

This paper studies problems where data is continuously collected from a constantly evolving distribution such that the single train/test paradigm no longer applies and the model learns and makes predictions based on already seen data to predict accurately on test data from future where the learner does *not* have access to any training data from future. Our main contribution is to introduce a simple yet effective time-varying propensity score estimator to account for gradual shifts in data distributions. One important property of our method is that it can be utilized in many different kinds of settings—from supervised learning to reinforcement learning—and can be used in tandem with many existing approaches for these problems. Another key property of our method is that it neither has any assumption on data nor requires data to be generated from a specific distribution in order to be effective. We evaluate our proposed method in various settings involving a sequence of gradually changing tasks with slow repeating patterns. In such settings, the model not only must continuously adapt to changes in the environment/task but also learn how to select past data which may have become more relevant for the current task. Extensive experiments show our method efficiently detects data shifts and statistically account for such changes during learning and inference time.

## 2 APPROACH

For two probability densities $p(x)$ and $q(x)$ that are absolutely continuous with respect to the Lebesgue measure on $\mathbb{R}^d$, observe that

$$\mathbb{E}_{x \sim q}[\ell(x)] = \int \frac{\mathrm{d}p(x)}{\mathrm{d}p(x)} \, \mathrm{d}q\,(x)\ell(x) = \mathbb{E}_{x \sim p}\left[\frac{\mathrm{d}q(x)}{\mathrm{d}p(x)}\ell(x)\right] = \mathbb{E}_{x \sim p}\left[\beta(x)\,\ell(x)\right] \tag{1}$$

where $\ell(x)$ is any function. The propensity score $\beta(x) = \frac{\mathrm{d}q(x)}{\mathrm{d}p(x)}$ is the Radon-Nikodym derivative between the two densities (Resnick, 2013). It measures the relative likelihood of a datum $x$ coming from $p$ versus $q$; in the sequel we will call $\beta(x)$ the "standard propensity score". We may think of $q(x)$ as the probability density corresponding to the test distribution and $p(x)$ as that of the training distribution. If $\ell(x)$ is the loss of a datum, the identity here suggests that we can optimize the test loss, i.e., the left-hand-side, using a $\beta$-weighted version of the training loss, i.e., the right-hand-side. In practice, can estimate $\beta(x)$ via samples from $p$ and $q$ (Agarwal et al., 2011) as follows. We can create a data set $D = \{(x_i, z_i)\}_{i=1}^{N}$ with $z_i = 1$ if $x_i \sim p$ and $z_i = -1$ if $x_i \sim q$. We fit a model $g_\theta$, which can be linear or non-linear, parameterized by $\theta$ to classify these samples and obtain

$$\hat{\theta} = -\min_{\theta} \frac{1}{N} \sum_{i=1}^{N} \log\left(1 + \exp(-z_i g_\theta(x_i))\right). \tag{2}$$

### 2.1 MODELING AND ADDRESSING DRIFT IN THE DATA

Let us now consider the situation when the probability density evolves over time: $(p_t(x))_{t \leq T}$ where $t \in \mathbb{R}$. Assume that we have a dataset $D = \{(x_i, t_i)\}_{i=1}^{N}$ where each datum $x_i$ was received at time $t_i$ and therefore $x_i \sim p_{t_i}(x)$. The dataset can contain multiple samples from any of the $p_t$ for $t \leq T$, and there may even be time instants for which there are no samples. Our theoretical development will be able to address such situations. Let $p(t) = N^{-1} \sum_{i=1}^{N} \delta_{t_i}(t)$ be the empirical distribution of the time instants at which we received samples; here $\delta$ denotes the Dirac-delta distribution. We can now define the marginal on the data as

$$p(x) = \int_0^T \mathrm{d}t \, p(t)\, p_t(x).$$

Inspired by this expression, we will liken $p_t(x) \equiv p(x \mid t)$.

**Assumption 1 (Data evolves gradually).** The probability density of data at a later time

$$p_t(x); t > T$$

can be arbitrarily different from the past data and in the absence of prior knowledge of how the distribution evolves, we cannot predict accurately far into the future. Suppose our goal is to build a

model that can predict well on data from $p_{T+dt}(x)$, i.e., a small time-step into the future using the dataset $D$. For the loss $\ell(x)$, this amounts to minimizing

$$\mathbb{E}_{x \sim p_{T+dt}} [\ell(x)]. \tag{3}$$

This problem is challenging because we do not have any data from $p_{T+dt}$, at training time or for adaptation. But observe $\mathbb{E}_{x \sim p_{T+dt}(x)} [\ell(x)]$ equals to

$$\mathbb{E}_{x \sim p_T(x)} [\ell(x)] + \int dx \, (p_{T+dt}(x) - p_T(x)) \, \ell(x) \leq \mathbb{E}_{x \sim p_T(x)} [\ell(x)] + 2\mathrm{TV}(p_{T+dt}(x), p_T(x)), \tag{4}$$

where $\mathrm{TV}(p_{T+dt}, p_T) = \frac{1}{2} \int dx \, |p_{T+dt}(x) - p_T(x)|$ is the total variational divergence between the probability densities at time $T$ and $T + dt$, and the integrand $\ell(x)$ is upper bounded by 1 in magnitude (without loss of generality, say after normalizing it). We therefore assume that the changes in the probability density $p_T$ are upper bounded by a constant (uniformly over time $t$); this can allow us to build a model for $p_{T+dt}$ using data from $(p_t)_{t \leq T}$.

**Validating and formalizing Assumption 1** This assumption is a natural one in many settings and we can check it in practice using a two-sample test to estimate the total variation $\mathrm{TV}(p_{T+dt}, p_T)$ (Gretton et al., 2012). We do so for some of our tasks in Sec. 3 (see Fig. 9). We can also formalize this assumption mathematically as follows. Consider a stochastic process $(X_t)_{t \in \mathbb{R}}$ with a transition function $P_t$. By definition of the transition function, we have that

$$\mathbb{E} [\varphi(X_t)] = \int \varphi(y) P_t(X_s, dy) \quad \forall s \leq t,$$

for any measurable function $\varphi$. The distribution of the random variable $X_t$ corresponding to this Markov process is exactly the distribution of data $p_t$ at time instant $t$ in our formulation. We can now define the semi-group of this transition function as the conditional expectation $K_t \varphi = \int dy \, \varphi(y) P_t(x, dy)$; this holds for any function $\varphi$. Such a semi-group satisfies properties that are familiar from Markov chains in finite state-spaces, e.g., $K_{t+s} = K_t K_s$. The infinitesimal generator of the semi-group $K_t$ is defined as $A\varphi = \lim_{t \to 0} (\varphi - K_t \varphi)/t$. This generator completely defines the local evolution of a Markov process (Feller, 2008)

$$p_t = e^{-tA} \, p_0 \tag{5}$$

where $e^{tA}$ is the matrix exponential of the generator $A$. If all eigenvalues of $A$ are positive, then the Markov process has a steady state distribution, i.e., the distribution of our data $p_t$ stops evolving (this could happen at very large times and the time-scale depends upon the smallest non-zero eigenvalue of $A$). On the other hand, if the magnitude of the eigenvalues of $A$ is upper bounded by a small constant, then the stochastic process evolves gradually.

The above discussion suggests that our development in this paper, building upon Assumption 1, is meaningful, so long as the data does not change abruptly.

**Remark 2 (Evaluating the model learned from data from $(p_t)_{t \leq T}$ on test from $p_{T+dt}$).** We are interested in making predictions on future data, i.e., data from $p_{T+dt}$. For all our experiments, we will therefore evaluate on test data from "one time-step in the future". The learner does not have access to any training data from $p_{T+dt}$ in our experiments. This is reasonable if the data evolves gradually. Our setting is therefore different from typical implementations of online meta-learning (Finn et al., 2019; Harrison et al., 2020) and continual learning (Hadsell et al., 2020; Caccia et al., 2020).

In the sequel, motivated by Assumption 1, we will build a time-dependent propensity score for $p_T$ instead of $p_{T+dt}$. Using a similar calculation as that of Eq. (1) (see Appendix A for a derivation), we can write our objective as

$$\mathbb{E}_{x \sim p_T(x)} [\ell(x)] = \mathbb{E}_{t \sim p(t)} \mathbb{E}_{x \sim p_t(x)} [\omega(x, T, t) \, \ell(x)], \tag{6}$$

where

$$\omega(x, T, t) = \frac{dp_T}{dp_t}(x) \tag{7}$$

is the time-varying propensity score.

**Remark 3 (Is the time-varying propensity score equivalent to standard propensity score on $(x, t)$?).** We can define a new random variable $z \equiv (x, t)$ with a corresponding distribution $p(z) \equiv p_t(x)p(t)$. However, doing so is not useful for estimating $\mathbb{E}_{x \sim p_T(x)} [\ell(x)]$ because our objective Eq. (6) involves conditioning on a particular time $T$ (see also Sec. 3.3 for numerical comparison).

## 2.2 TIME-VARYING PROPENSITY SCORE

Let us first discuss a simple situation where we model the time-varying propensity score using an exponential family (Pitman, 1936; Wainwright & Jordan, 2008)

$$p_t(x) = p_0(x) \exp\left(g_\theta(x, t)\right) \tag{8}$$

where $g_\theta(x, t)$ is a continuous function parameterized by weights $\theta$. This gives

$$\omega(x, T, t) = \exp\left(g_\theta(x, T) - g_\theta(x, t)\right) \tag{9}$$

The left-hand-side can be calculated using Eq. (2) to estimate $g_\theta(x, t)$ as follows. We create a modified dataset whose inputs are triplets $(x_i, t', t)$ with a label 1 if $t' = t_i$, i.e., the time at which the corresponding $x_i$ was recorded, and a label -1 if $t = t_i$.

**Remark 4 (Modeling the propensity score using an exponential family does not mean that the data is from an exponential family).** Observe that the exponential family in Eq. (8) is used to model the propensity $p_t(x)/p_0(x)$, this is different from $p_t(x) \propto \exp(g_\theta(x, t))$ which would amount to modeling the data distribution as belong to an exponential family.

Observe that in the above method $p_0(x)$, which is the distribution of data at the initial time, is essentially unconstrained. We can exploit this "gauge" to choose $p_0(x)$ differently and build a more refined estimator for the time-varying propensity score. We can model $p_t(x)$ as deviations from the marginal on $x$:

$$p_t(x) = p(x) \exp\left(g_\theta(x, t)\right). \tag{10}$$

This gives

$$\omega(x, T, t) = \exp\left(g_\theta(x, T) - g_\theta(x, t)\right).$$

In this case, we create a modified dataset where for each of $N$ samples, there is an input tuple $(x_i, t)$ with label 1 if $t = t_i$, i.e., the correct time at which $x_i$ was recorded, and with label -1 if $t \neq t_i$. While creating this modified dataset, we choose all the unique time instances at which some sample was recorded, i.e., $t \in \{t_i : (x_i, t_i) \in D\}$. The logistic regression in Eq. (2) is fitted to obtain $g_\theta(x, t)$ using this modified dataset. Algorithm 2 gives more details.

It is important to note that the distinction between the two approaches is subtle. The Eq. (8) models the deviations of the probability density $p_t(x)$ from its value at the first time instant $p_0(x)$ while Eq. (10) models its deviations from the marginal $p(x) = \int_0^T \mathrm{d}t\, p(t)p_t(x)$. The nuances between these approaches are subtle when deals with data with varying gradual shift, leading to comparable performance of both methods, as illustrated in Fig. 8.

## 2.3 A LEARNING-THEORETIC PERSPECTIVE

We next discuss a learning theoretic perspective on how we should learn models when the underlying data evolves with time. For the purposes of this section, suppose we have $m$ input-output pairs $D_t = \{(x_i, y_i)\}_{i=1}^m$ where $x_i$ are the inputs and $y_i \in \{0, 1\}$ are the binary outputs, that are independent and identically distributed from $p_t(x, y)$ for $t = 1, \ldots, T$. Note that data from each time instant are IID, there can be correlations across time. Unlike the preceding sections when time was real-valued, we will only consider integer-valued time.

Consider the setting where we seek to learn hypotheses $\overline{h} = (h_t)_{t=1,\ldots,T} \in H^T$ that can generalize on new samples from distributions $(p_t)_{t=1,\ldots,T}$ respectively. A uniform convergence bound on the population risk of one $h_t$ suggests that if

$$m = \mathcal{O}\left(\epsilon^{-2}(\mathrm{VC}_H - \log \delta)\right),$$

then $e_t(h_t) \leq \hat{e}_t(h_t) + \epsilon$ with probability at least $1 - \delta$; here $e_t(h_t)$ the population risk of $h_t$ on $p_t$, $\hat{e}_t(h_t)$ is its average empirical risk on the $m$ samples from $D_t$ (Vapnik, 1998) and VC is the VC-dimension of $H \ni h_t$. If we want hypotheses that have a small average population risk $e^t(\overline{h}) = T^{-1} \sum_{t=1}^T e_t(h_t)$. across all time instants, we may minimize the empirical risk $\hat{e}^t(\overline{h}) = T^{-1} \sum_{t=1}^T \hat{e}_t(h_t)$. As Baxter (2000) shows, if

$$m = \mathcal{O}\left(\epsilon^{-2}\left(\mathrm{VC}_H(T) - T^{-1}\log \delta\right)\right) \tag{11}$$

then we have $e^t(\overline{h}) \leq \hat{e}^t(\overline{h}) + \epsilon$ for any $\overline{h}$. The quantity $\text{VC}_H(T)$ is a generalized VC-dimension that characterizes the number of distinct predictions made by the $T$ different hypotheses; it depends upon the stochastic process underlying our individual distributions $p_t$. Larger the time $T$, smaller the $\text{VC}_H(T)$. Whether we should use the $m$ samples from $p_T$ to train one hypothesis, or use all the $mT$ samples to achieve a small *average* population risk across all the tasks, depends upon how related the tasks are. If our goal is only the latter, then we can just train on data from all the tasks because

$$\text{VC}_H(T) \leq \text{VC}_H, \text{ for all } T \geq 1.$$

In the literature on multi-task or meta-learning, this result is often the motivation for learning on samples from all the tasks. We will use this as a baseline in our experiments; we call this **Everything** in Sec. 3.1.1. In theory, this baseline is effective if tasks are related to each other (Baxter, 2000; Gao & Chaudhari, 2021; Achille et al., 2019) but it will not be able to exploit the fact that the data distribution evolves over time. A related baseline that we can glean from this argument that the one that only builds the hypothesis using data from $p_T$; we call this **Recent**) in Sec. 3.1.1.

Our goal is however not to obtain a small population risk on all tasks, it is instead to obtain a small risk on $p_{T+dt}$—our proxy for it being the latest task $p_T$. Even if the average risk on all tasks of the hypotheses trained above is small, the risk on a particular task $p_T$ can be large. (Ben-David & Schuller, 2003) studies this phenomena. They show that if we can find a hypothesis $h \circ f_t$ such that $h \circ f_t$ can predict accurately on $p_t$ for all $t$, then this effectively reduces the size of the hypothesis space $\overline{h} \in H^T$ and thereby entails a smaller sample complexity in Eq. (11) at the added cost of searching for the best hypothesis $f_t \in F$ for each task $p_t$ (which requires additional samples that scale linearly with $\text{VC}_F$). The sample complexity of such a two-stage search can be better than training on all tasks, or training on $p_T$ in isolation, under certain cases, e.g., if $\text{VC}_H \geq \log |F|$ for a finite hypothesis space $F$ (Ben-David & Schuller, 2003).

While the procedure that combines such hypotheses to get $h \circ f_t$ in the above theory is difficult to implement in practice, this argument motivates our third baseline (**Finetune**) which first fits a hypothesis on all past tasks $(p_t)_{t \leq T}$ and then adapts it further using data from $p_T$.

## 3 EXPERIMENTS

We provide a broad empirical comparison of our proposed method in both continuous supervised and reinforcement learning. We first evaluate our method on synthetic data sets with time-varying shifts. Next, we create continuous supervised image classification tasks utilizing CIFAR-10 (Krizhevsky & Hinton, 2009) and CLEAR (Lin et al., 2021) datasets. Finally, we evaluate our method on ROBEL D'Claw (Ahn et al., 2019; Yang et al., 2021) simulated robotic environments and MuJoCo (Todorov et al., 2012). See also Appendices C and D for a description of our settings and more results.

### 3.1 CONTINUOUS SUPERVISED LEARNING

In these experiments, the goal is to build a model that can accurately predict future data (under continuous label shift) at $t + 1$ using *only* historical data from $\{p_0, \ldots, p_t\}$ as described in Remark 2. It is important to note that all models, including baselines, are trained *entirely from scratch* at each time step. Also, we do not update the model continually, as our main focus is on understanding the the performance of the time-varying propensity score. The following sections will explain the baseline models, the data sets and simulated shift setups, and the results of the experiments.

#### 3.1.1 BASELINE METHODS

**1. Everything:** To obtain model used at time $t$, we train on pooled data from all past times $\{p_s(x) : s \leq t\}$, including the most recent data from $p_t(x)$. More precisely, this involves minimizing the objective

$$\ell^t(\phi) = \frac{1}{N} \sum_{s=1}^{t} \sum_{\{x_i : t_i = s\}} \ell(\phi; x_i). \tag{12}$$

where the loss of the predictions of a model parameterized by weights $\phi$ is $\ell(\phi; x_i)$ and we have defined $\ell^t(\phi)$ as the cumulative loss of all tasks up to time $t$.

**2. Recent** trains only on the most recent data, i.e., data from $p_t(x)$. This involves optimizing

$$\ell_t(\phi) = \frac{1}{N'} \sum_{\{x_i:t_i=t\}} \ell(\phi;x_i), \text{ where } N' = \sum_{\{x_i:t_i=t\}} 1. \tag{13}$$

**3. Fine-tune:** first trains on all data from the past using the objective

$$\phi^{t^-} = \arg\min_{\phi} \frac{1}{N} \sum_{s=1}^{t} \sum_{\{x_i:t_i=s,s<t\}} \ell(\phi;x_i)$$

and then finetunes this model using data from $p_t$. We can write this problem as

$$\ell^{t^+}(\phi) = \frac{1}{N'} \sum_{\{x_i:t_i=t\}} \ell(\phi;x_i) + \Omega(\phi - \phi^{t^-}), \tag{14}$$

where $\Omega(\phi - \phi^{t^-})$ is a penalty that keeps $\phi$ close to $\phi^{t^-}$, e.g., $\Omega(\phi - \phi^{t^-}) = \|\phi - \phi^{t^-}\|_2^2$. Sometimes this approach is called "standard online learning" (Finn et al., 2019) and it resembles (but is not the same as) Follow-The-Leader (FTL) (Shalev-Shwartz, 2012). We call it "Finetune" to explicitly state the objective and avoid any confusion due to its subtle differences with FTL. In practice, we implement finetuning without the penalty $\Omega$ by initializing the weights of the model to $\phi^{t^-}$ in Eq. (14).

**Remark 5 (Properties of the different baselines).** Recall that our goal is to build a model that predicts on data from $p_{t+1}(x)$. It is important to observe that **Everything** minimizes the objective over all the past data; for data that evolves over time, doing so may be detrimental to performance at time $t$. If we think using a bias-variance tradeoff, the variance of the predictions of **Everything** is expected to be smaller than the variance of predictions made by **Recent**, although the former is likely to have a larger bias. For example, if the data shifts rapidly, then we should expect **Everything** to have a large bias and potentially a worse accuracy than that of **Recent**. But if the amplitude of the variations in the data is small (and drift patterns repeat) then **Everything** is likely to benefit from the reduced variance and perform better than **Recent**. **Fine-tune** strikes a balance between the two methods and although the finetuning procedure is not always easy to implement optimally, e.g., the finetuning hyper-parameters can be different for different times. As our results illustrate neither of these approaches show consistent trend in the experiments and highly depend on a given scenario which can be problematic in practice, e.g. sometimes **Everything** performs better than both but other times **Fine-tune** is better (compare their performance in Fig. 1).

**Remark 6 (How does the objective change when we have supervised learning or reinforcement learning?).** We have written Eqs. (12) to (14) using the loss $\ell(\phi;x_i)$ only for the sake of clarity. In general, the loss depends upon both the inputs $x_i$ and their corresponding ground-truth labels $y_i$ for classification problems. For problems in reinforcement learning, we can think of our data $x_i$ as entire trajectories from the system and thereby the loss $\ell(\phi;x_i)$ as the standard 1-step temporal difference (TD) error which depends on the weights that parameterize the value function.

We incorporate our time-varying propensity score into Eq. (12) as follows:

$$\ell_\omega^t(\phi) = \frac{1}{N} \sum_{s=1}^{t} \sum_{\{x_i:t_i=s\}} \boldsymbol{\omega}(x_i,t,t_i) \, \ell(\phi;x_i). \tag{15}$$

Note that the *only* difference between our method and **Everything** is introduction of $\boldsymbol{\omega}$ into Eq. (12) and all other details (e.g. network architecture, etc.) are exactly the same. $\boldsymbol{\omega}$ can be trained based on our algorithm explained in Sec. 2.1 and Algorithm 2. Important to note that, our method automatically detects shifts in the data without knowing them as a priori. This means that our method is able to detect these changes without being given any explicit information about when or where they occur.

### 3.1.2 DRIFTING GAUSSIAN DATA

Here, we design continuous classification tasks using a periodic shift where samples are generated from a Gaussian distribution with time-varying means ($\mu_t$) and standard deviation of 1. In particular, we create a data set, $D^t = \{(x_i^t, y_i^t)\}_{i=1}^{N}$, at each time step $t$ where $x_i^t \sim \mathcal{N}(\mu_t, 1), \mu_t = \mu_{t-1} + d/10,$

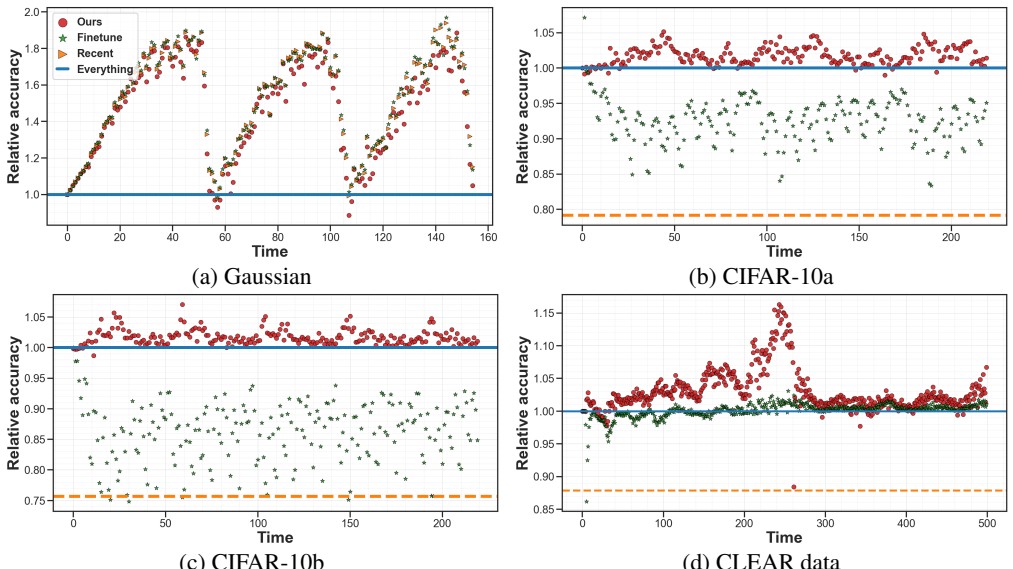

Figure 1: **Relative test accuracy achieved by our method and others vs Everything in continuous supervised learning benchmarks**. The x-axes indicate results on test data at each time step $t$ as described in Sec. 3.1.3, and points above the line (———) indicate better performance than **Everything**. We see that our method stands out as the only approach that consistently maintains its effectiveness regardless of the specific setting or the diversity of the past data. It is important to note that all models in these experiments are trained entirely from scratch per time step and each point on these plots represents a *different* model resulted in large number of experiments (e.g. Fig. 1d shows 500 experiments per method). To enhance the clarity, we only display the mean accuracy achieved by the **Recent** using a dashed line, as it performs the worst in the image benchmarks.

$\mu_0 = 0.5$, and label 1 ($y_i^t = 1$) is assigned if $x_i^t > \mu_t$, otherwise $y_i^t = 0$. We change direction of shift every 50 time steps (i.e., set $d \leftarrow -d$) and use $N = 2000$ in our experiments. We run these experiments for 160 time steps. Fig. 1a displays that our method performs much better than **Everything** and similarly to others. We note that both **Recent** and **Fine-tune** are like oracle for this experiment as test time data from $p_{t+1}(x)$ are more similar to the recent data than other historical data. This experiment clearly illustrates how our method enables model training to selectively emphasize data similar to the current time and, importantly, data that evolved in a similar fashion in the past.

### 3.1.3 IMAGE CLASSIFICATION WITH EVOLVING LABELS

We adopt the experimental setting of (Wu et al., 2021) for classification under continuous label shift with images from CIFAR-10 (Krizhevsky & Hinton, 2009) and CLEAR (Lin et al., 2021).

**CIFAR-10.** We split the original CIFAR-10 training data into 90% (train) and 10% (validation), and use original test split as is. Following (Wu et al., 2021), we pre-train Resent-18 (He et al., 2016) using training data for 100 epochs. In subsequent continuous label shift experiments, we utilize the pre-trained model and only fine-tune its final convolution block and classification (output) layer. We use two different shifts, called CIFAR-10a and CIFAR-10b (see Appendix D.1 for details) and 3 random data splits here.

**CLEAR.** The data set contains 10 classes with 3000 samples per class. CLEAR is curated from a large public data set spanning 2004 to 2014. We utilize unsupervised pre-trained features provided by (Lin et al., 2021) and only use a linear layer classifier during continuous label shift experiments. As they stated, the linear model using pre-trained features is far more effective than training from scratch for this data set (for more, see (Lin et al., 2021)). We average all results over 3 random train-validation-test splits, of relative size 60% (train), 10% (validation), and 30% (test).

**Continuous label shifts.** We create a sequence of classification tasks wherein the label distributions varies at each time step, according to a predefined pattern (refer to Appendix D.1 for details). The pattern repeats slowly, with each new task having a slightly different label distribution than previous task. Here, we train a model from *scratch* at each time step $t$ using all training data encountered thus

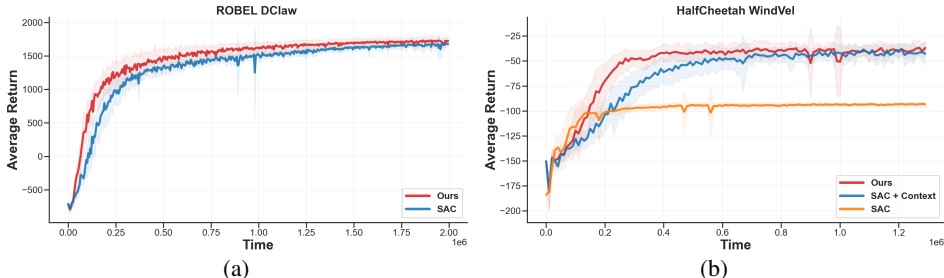

Figure 2: **Comparison of the average undiscounted return (higher is better) of our method (red) against baseline algorithms on ROBEL D'Claw and Half-Cheetah WindVel environments**. Our method consistently outperforms other methods in terms of sample complexity across all environments.

far and evaluate the model on test data drawn from label distribution of time $t + 1$. This resulted in a significant number of experiments, typically ranging from 100 to 500 different models per benchmark illustrated in Fig. 1. This setup presents a significant challenging scenario due to the occurrence of label shift at each time step. Consequently, it becomes crucial for a method to learn how to selectively utilize only data which are relevant for the current time step and task. It is important to note that we solely employ the test data during the evaluation phase and never use it to train a model.

**Results.** Fig. 1 shows the results of these large experiments, where we compare our method against the baselines across different data sets and shifts. To give further insight into our method, we show in Fig. 9 that our method produces high quality estimates under shifted distribution. Also, in Fig. 4, we present a comparison between our method and Meta-Weight-Net (Shu et al., 2019). All these comprehensive results illustrate that not only does our method outperform others by achieving higher classification accuracy, but it also consistently performs well across various shifts and benchmarks, unlike other methods. For instance, **Everything** fares poorly in Gaussian experiment than others but it works better than **Fine-tune** and **Recent** on continuous CIFAR-10 benchmark. We also show the comparable performance of our method and **Everything** in the absence of data shifts, highlighting their consistent applicability and effectiveness (see Appendix C for more).

## 3.2 REINFORCEMENT LEARNING WITH CHANGING TASKS

In what follows, we discuss how we adopt our method into existing RL methods. We use two challenging environments with various settings for experiments in this section. For all experiments, we report the undiscounted return averaged across 10 different random seeds.

**ROBEL D'Claw.** This is a robotic simulated manipulation environment containing 10 different valve rotation tasks and the goal is to turn the valve below 180 degrees in each task. All tasks have the same state and action spaces, and same episode length but each rotates a valve with a different shape. We build a random sequence of these tasks switching from one to another (after an episode finishes) in repeating patterns. We utilize SAC (Haarnoja et al., 2018) as our baseline. In order to incorporate our approach into SAC, we only change Q-value update with $\boldsymbol{\omega}$ as follows:

$$\frac{1}{|B|} \sum_{(s,a,t,s')} \left[ \boldsymbol{\omega}(s,a,T,t) \Big( Q_\psi(s,a) - r - \gamma Q_{\hat{\psi}}(s',a') + \alpha \log \pi_\phi(a'|s') \Big)^2 \right], a' \sim \pi_\phi(\cdot|s') \quad (16)$$

where $\alpha$ is entropy coefficient, $T$ is the current episode, and $(s,a)$ denote state and action, respectively.

**Half-Cheetah WindVel.** We closely follow the setup of (Xie et al., 2020) to create *Half-Cheetah WindVel* environment where direction and magnitude of wind forces on an agent keep changing but slowly and gradually. This is a difficult problem as an agent needs to learn a non-stationary sequence of tasks (i.e. whenever direction and magnitude of wind changes, it constitutes as a new task). Since task information is not directly available to the agent, we utilize the recent method of Caccia et al. (2022) and Fakoor et al. (2020c) to learn and infer task related information using context variable (called SAC + Context here). Similar to Eq. (16), we only change Q-update to add our method. We emphasize that SAC + Context is a strong method for non-stationary settings and leads to state-of-art performance in those scenarios (Fakoor et al., 2020c; Ni et al., 2022). Additionally, we use SAC in this experiment as well to underscore the challenging nature of the task.

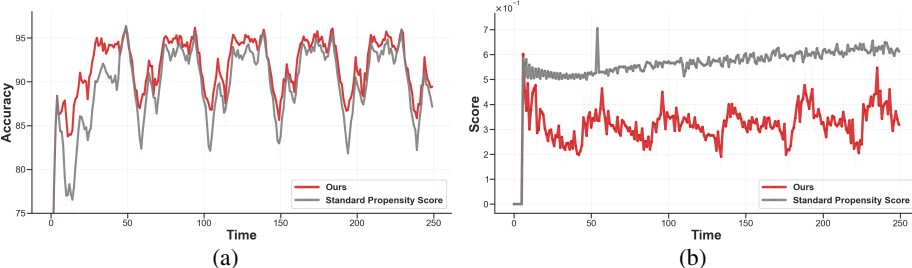

Figure 3: **Comparing our method against standard propensity on continuous CIFAR-10 benchmark**. The x-axis indicates results on test data at each time step. Fig 3a shows classification accuracy and Fig 3b shows propensity scores. These results clearly demonstrate that our time-varying propensity score appropriately weighs the data as it evolves, providing a plausible explanation for the better performance of our method compared to the standard propensity score. The standard propensity score is unable to identify the evolution of the data, which hinders its effectiveness in adapting to changing conditions.

**Results.** We can see in Fig. 2 that our method offers consistent improvements over the baseline methods. This is particularly noteworthy that although the replay buffer contains data from various behavior policies, our method can effectively model how data evolve over time with respect to the current policy *without* requiring to have access to the behavior policies that were used to collect the data. It is particularly useful in situations where the data are often collected from unknown policies. Note that the only difference between our method and baseline methods is the introduction of $\omega$ term in the Q-update and all other details are exactly the same. See Fig. 10 and 11 for more results.

### 3.3 COMPARING STANDARD PROPENSITY WITH OUR METHOD

Now we compare our method with standard propensity scoring discussed in Sec. 2. Fig. 3 shows results of this experiment on CIFAR-10b benchmark. For a comparison with the Gaussian benchmark, please refer to Fig. 5. These result provide an empirical verification that our method works better than standard propensity method and gives us some insights why it performs better. Particularly, we can see in Fig. 3 and 5 that our method fairly detects shifts across different time steps, whereas standard propensity scoring largely ignores shifts across different time steps. Moreover, one major issue with standard propensity is which portion of the data is considered current and which is considered outdated when creating a binary classifier as Eq. (2). This challenge becomes more complex when data is continuously changing, and what was considered current in one step becomes old in the next. However, our method does not suffer from this issue as it organically depends on time.

## 4 DISCUSSION

We propose a straightforward yet powerful approach for addressing gradual shifts in data distributions: a time-varying propensity score. Much past work has focused on a single shift between training/test data (Lu et al., 2021; Wang & Deng, 2018; Fakoor et al., 2020c; Chen et al., 2016) as well as restricted forms of shift involving changes in only the features (Sugiyama et al., 2007a; Reddi et al., 2015a), labels (Lipton et al., 2018; Garg et al., 2020; Alexandari et al., 2020), or in the underlying relationship between the two (Zhang et al., 2013; Lu et al., 2018). Past approaches to handle distributions evolving over time have been considered in the literature on: concept drift (Gomes et al., 2019; Souza et al., 2020), survival analysis (Cox, 1972; Lu, 2005), reinforcement learning (shift between the target policy and behavior policy) (Schulman et al., 2015; Wang et al., 2016; Fakoor et al., 2020a;b; 2021), (meta) online learning (Shalev-Shwartz, 2012; Finn et al., 2019; Harrison et al., 2020; Wu et al., 2021), and task-free continual/incremental learning (Aljundi et al., 2019; He et al., 2019). However, to best our knowledge, existing methods for these settings do not employ time-varying propensity weights like we propose here. One of the key advantages of our method is its ability to automatically detect data shifts without prior knowledge of their occurrence. Additionally, our method is versatile and can be applied in various problem domains, including supervised learning and reinforcement learning and it can be seamlessly integrated with existing approaches. Through extensive experiments in continuous supervised learning and reinforcement learning, we demonstrate the effectiveness of our method. It is important to note that while we refer to "time" in this paper, our methodology can also be applied to situations where shifts occur gradually over space or any other continuous-valued index. The broader impact of this research lies in enhancing the robustness of machine learning techniques in real-world scenarios that involve continuous distribution shifts.

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

# Appendix: Time-Varying Propensity Score to Bridge the Gap between the Past and Present

## A  DERIVATIONS

Using a similar calculation as that of Eq. (1), we can write our time-varying propensity score-based objective as follows

$$\mathbb{E}_{x \sim p_T(x)}\Big[\ell(x)\Big] = \mathbb{E}_{x \sim p(x|T)}\Big[\ell(x)\Big] = \int_t \mathrm{d}p(x|T)\,\ell(x) = \int_t \int_x \mathrm{d}p(t)\,\mathrm{d}p(x|T)\,\frac{\mathrm{d}p(x|t)}{\mathrm{d}p(x|t)}\ell(x) \quad (17)$$

$$= \int_t \int_x \mathrm{d}p(t)\,\mathrm{d}p(x|t)\,\frac{\mathrm{d}p(x|T)}{\mathrm{d}p(x|t)}\ell(x)$$

$$= \int_t \mathrm{d}p(t) \int_x \mathrm{d}p(x|t)\,\frac{\mathrm{d}p(x|T)}{\mathrm{d}p(x|t)}\ell(x)$$

$$= \mathbb{E}_{t \sim p(t)}\mathbb{E}_{x \sim p(x|t)}\Big[\frac{\mathrm{d}p(x|T)}{\mathrm{d}p(x|t)}\ell(x)\Big]$$

$$= \mathbb{E}_{t \sim p(t)}\mathbb{E}_{x \sim p_t(x)}\Big[\frac{\mathrm{d}p_T(x)}{\mathrm{d}p_t(x)}\ell(x)\Big]$$

$$= \mathbb{E}_{t \sim p(t)}\mathbb{E}_{x \sim p_t(x)}\Big[\boldsymbol{\omega}(x, T, t)\ell(x)\Big] \quad (18)$$

$\boldsymbol{\omega}$ can be parameterized by a neural network and is trained based on our algorithm explained in the Appendix B.

## B  ALGORITHM DETAILS

We learn $\boldsymbol{\omega}$ by building a binary classifier by utilizing samples from different time steps. In particular, we need to create a triplet $(x_i, t', t)$ for each of samples in the dataset with label 1 if $t'$ equals the time corresponding to $x_i$, i.e., if $t_i = t'$ and label $-1$ if $t$ equals the time corresponding to $x_i$, i.e. $t_i = t$. These steps are detailed in Algorithm 1.

---

**Algorithm 1** $GenerateData$

---

1: **Input**: $\mathcal{D} = \{x_j, t_j\}_{j=1}^N$
2: $\mathcal{T} = [1, .., T]$
3: Initialize $\mathcal{D}_o = \varnothing$
4: **for** j=1...N **do**
5:     $t_r$ is chosen uniformly at random from $\mathcal{T}$ such that $t_j \neq t_r$
6:     **if** random() $>= 0.5$ **then**
7:         $z = 1$
8:         $\mathcal{D}_o \leftarrow \mathcal{D}_o \cup \{(x_j, \mathbf{t_j}, t_r, z)\}$
9:     **else**
10:         $z = -1$
11:         $\mathcal{D}_o \leftarrow \mathcal{D}_o \cup \{(x_j, t_r, \mathbf{t_j}, z)\}$
12:     **end if**
13: **end for**
14: **Return** $\mathcal{D}_o$

---

Once the triplet $(x_i, t', t)$ are generated, we fit the binary classifier by solving the following:

$$- \underset{\theta}{\text{minimize}} \; \frac{1}{N} \sum_{(x, t_2, t_1, z) \in \mathcal{B}}^{N} \log\Big(1 + e^{-z(g_\theta(x, t_2) - g_\theta(x, t_1))}\Big) \quad (19)$$

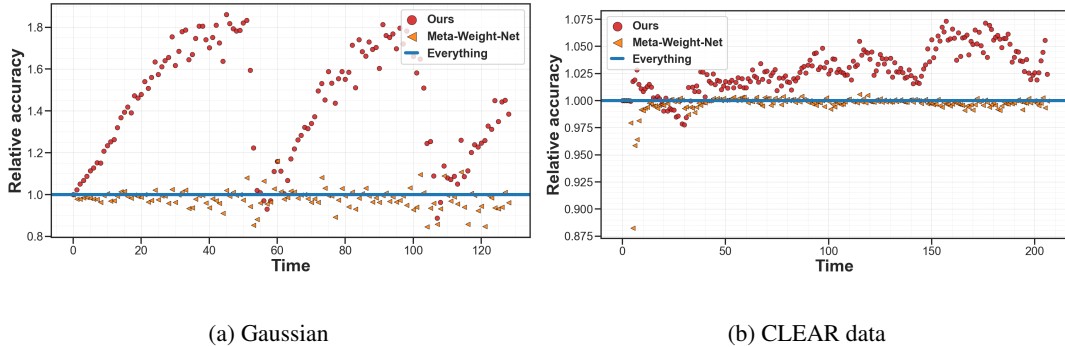

(a) Gaussian

(b) CLEAR data

Figure 4: **Relative test accuracy achieved by our method and others vs Everything on continuous supervised learning benchmarks**. The x-axes indicate accuracy on test data at each time step $t$ as described in Sec. 3.1.3, and points above the line ( ⎯ ) indicate better accuracy than **Everything**. The setup of this experiment is exactly the same as that of Fig. 1; **Meta-Weight-Net** is added as another baseline. We see that our method is effective regardless of the specific setting or the diversity of the past data; in particular it is performs better than **Meta-Weight-Net**.

where $\omega(x, T, t) = \exp\left(g_\theta(x, T) - g_\theta(x, t)\right)$ indicates our time-varying propensity score described in Sec. 2.1. Algorithm 2 presents the detailed implementation steps of our method.

It should be noted that we use the *exact* same algorithm (Algorithm 2) to learn $\omega$ for both reinforcement learning and continuous supervised learning experiments. The main difference between these cases is using of $(x, y, t)$ as inputs for supervised learning tasks whereas $(s, a, t)$ are used as inputs in the reinforcement learning experiments. Here $s$, $a$, $x$, $y$, and $t$ represent state, action, input data, label, and time respectively. Note although $g_\theta$ can be parameterized differently considering its given task (e.g. image classification tasks require $g_\theta$ to be a convolutional deep network; however, continuous control tasks in reinforcement learning experiments need fully connected networks), its implementation details remain the same. Moreover, in order to implement Eq. (10) described in Sec. 2.1, we only need to change line 8 of Algorithm 2 as follow (all other details remain the same):

$$\nabla_\theta J \leftarrow -\nabla_\theta \sum_{(x, t_2, z) \in \mathcal{B}} \log\left(1 + e^{-z(g_\theta(x, t_2))}\right) \tag{20}$$

---

**Algorithm 2** Train Time-varying Propensity Score Estimator

---

1: **Input**: $\mathcal{D} = \{x_j, t_j\}_{j=1}^N$
2: **Input** $M$: Number of epochs
3: Initialize $\mathcal{D}_o = \varnothing$
4: **for** j=1...M **do**
5:     $\mathcal{D}_o \leftarrow GenerateData(D)$ (see Algorithm 1)
6:     **repeat**
7:         Sample mini-batch $\mathcal{B} = \{(x, t_2, t_1, z)\} \sim \mathcal{D}_o$
8:         $\nabla_\theta J \leftarrow \nabla_\theta \sum_{(x, t_2, t_1, z) \in \mathcal{B}} \log\left(1 + e^{-z(g_\theta(x, t_2) - g_\theta(x, t_1))}\right)$
9:         $\theta \leftarrow \theta - \alpha \nabla_\theta J$
10:    **until** convergence
11: **end for**

---

It is important to point out that although Eq. (20) is used for the implementation of Algorithm 2 in all experiments presented in this paper, either method could have been utilized as Fig. 8 illustrates that both method performs similarly.

## C    ABLATION STUDIES AND ADDITIONAL EXPERIMENTAL RESULTS

**Method 1 vs Method 2.**    We compare performance of our Method 1 Eq. (8) and Method 2 Eq. (10) on continuous CIFAR-10 and CLEAR benchmarks. We can see in Fig. 8 that both methods perform similarly, although Method 1 has a slight edge over Method 2. Note we use Method 2 for all the experiments in the paper.

**Zoomed-in version of Fig. 1.**    We provide a zoomed-in version of Fig. 1 in Fig. 7 where we also remove **Recent** and **Fine-tune** to make the plots less crowded. Note that the experiments in these plots are exactly the same as the ones in Fig. 1 but only different visualizations.

**No Shift.**    Here, we analyze how our method performs when there is no shift in data and see whether or not it performs as good as **Everything** (Eq. (12))? To answer this question, we run new experiments on the CIFAR-10 benchmark *without* shift. We build continuous classification tasks from this dataset where we used past data to predict future data points. This is the same setting as other supervised learning experiments in the paper except we do not apply any shift to the data. Results of this experiment are shown in Fig. 14. As this experiment shows, our method and **Everything** perform similarly when there is no shift in the data. This experiment provides further evidence about the applicability of our method and shows that our method works regardless of presence of shifts in the data and it has no negative effect on the performance.

**Number of samples.**    Our approach involves utilizing fixed number of samples (i.e. $N$ in Eq. (19)) to train our time-varying propensity score as explained in Algorithm 1. Here, we investigate the influence of the sample size on the overall performance of our approach. We conduct new experiments on the CIFAR-10 benchmark, where we vary the sample size to 50, 100, and 300, while keeping all other settings consistent with the other supervised learning experiments described in the paper. The results of this experiment can be observed in Fig. 13. These experiments demonstrate that our method consistently outperforms **Everything** across different sample sizes. This further validates the versatility and effectiveness of our approach in various settings and scenarios.

**Additional experiments with more shifts.**    In order to further validate the effectiveness of our method, we conduct additional experiments on CIFAR-10c (see Appendix D.1 for more details). The results of this experiment are shown in Figure Fig. 6. Additionally, we run more experiments for our reinforcement learning settings, and the results of these experiments are illustrated in Fig. 10 and 11.

**Number of seeds.**    To assess the sensitivity of supervised learning results to the number of seeds, we conduct supplementary experiments employing 8 different random seeds and compare them with the results obtained from 3 seeds. The outcomes presented in Fig. 12 maintain consistency with those derived from 3 seeds. It's worth noting that for all reinforcement learning experiments, we employ 10 seeds.

## D    EXPERIMENT DETAILS

**Implementation Details.**    The hyper-parameters, computing infrastructure, and libraries utilized in the experiments of this paper are presented in Table 2, 1, and Table 3, respectively. It is worth mentioning that we conducted a minimal random hyper-parameters search for the experiments in this paper and we mostly follow standard and readily available settings for the experiments whenever it is applicable. We intend to make the code for this paper publicly available upon its publication.

### D.1    CONTINUOUS LABEL SHIFT SETTINGS

In this paper, we simulate the continuous label shift process for image classification experiments by adopting the setting introduced in (Wu et al., 2021). However, we modify and adapt their settings to create more challenging scenarios for our experiments. In particular, in contrast to the setting of simulating label shifts for only two classes as done in their study, we extend our label shift simulation to encompass all classes, totaling 10 classes in this case:

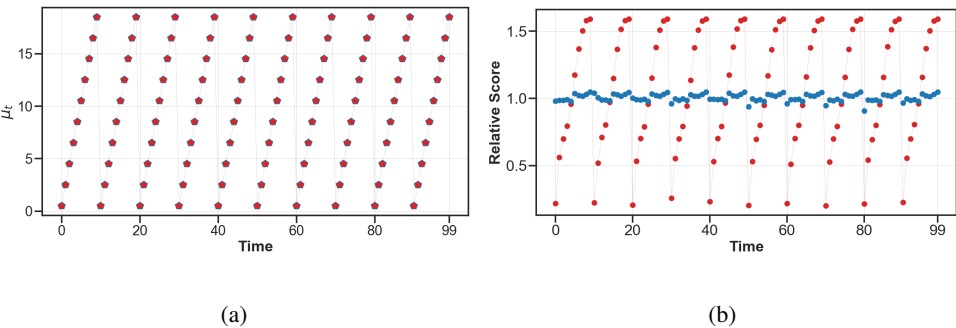

(a)                    (b)

Figure 5: **Comparing our method against standard propensity on Gaussian benchmark**. Fig 5a shows a periodic shift in which samples are generated from a Gaussian distribution with means ($\mu_t$) that vary over time; the standard deviation is 1 for all times. Fig 5b shows the score assigned by **standard propensity score** and **our** method. We have calculated a propensity score for each data point based on how similar it is to the samples at time 99. Data points that have a similar shift to the data point at time 99 should get a high score. For example, the data points at times 84, 79, 74, and 69 should have the highest scores, as they are the most similar to the data point at time 99. This experiment clearly demonstrates that our time-varying propensity score appropriately weighs the data as it evolves, providing a plausible explanation for the superior performance of our method compared to that of standard propensity score which is unable to identify the distribution shift which hinders its effectiveness in changing conditions. See Sec. 3.3 for more details.

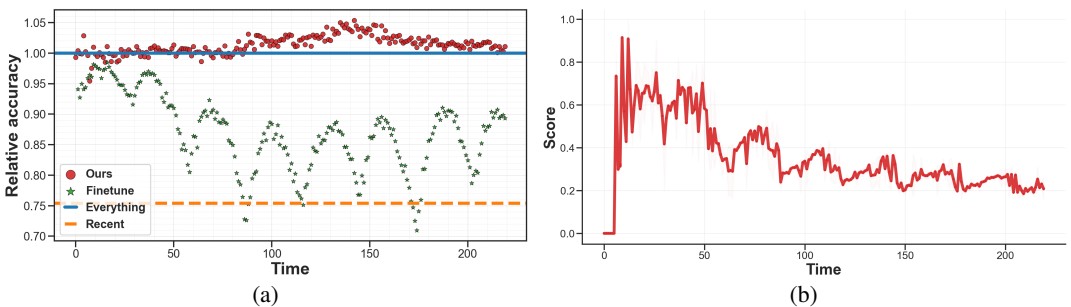

(a)                    (b)

Figure 6: **Relative test accuracy achieved by our method and others vs Everything on CIFAR-10c**. The x-axes indicate results on test data at each time step $t$ as described in Sec. 3.1.3, and points above the line (———) on the left indicate better performance than **Everything**. The y-axis on the right displays our time-varying propensity scores. These results once again highlight the effectiveness of our method compared to others. It is important to note that all models in these experiments are trained entirely from scratch per time step and each point on these plots represents a *different* model and experiment resulted in large number of experiment. To enhance the clarity, we only display the mean accuracy achieved by the **Recent** using a dashed line, as it performs the worst in the image benchmarks.

$$\forall i \in [1, C] \; \forall t \in [1, \mathcal{T}], v_t = (1 - \frac{t}{\mathcal{T}})q_t^i + (\frac{t}{\mathcal{T}})q_t^{i+1}, \tag{21}$$

where $\mathcal{T}$ is the number of shift steps per a label pair $i$ and $i + 1$, $t \in [1, \mathcal{T}]$, $C$ denotes the number of classes ($C = 10$ in our experiments), $q_t^i \in R^C$ is vector of class probabilities, and $v_t \in R^{\mathcal{T} \times C}$. For CIFAR-10 experiments, we use $\mathcal{T} = 9$ (called CIFAR-10a), $\mathcal{T} = 6$ (CIFAR-10b), and $\mathcal{T} = 30$ (CIFAR-10c). We also use $\mathcal{T} = 30$ for CLEAR experiment. Listing 1 demonstrates a simple Python implementation of the equation presented in Eq. (21).

```python
import numpy as np
def create_shift(T, q1, q2):
lamb = 1.0 / (T-1)
return np.concatenate([np.expand_dims(q1 * (1 - lamb * t) + q2 * lamb * t
    , axis=0)for t in range(T)], axis=0)
```

Listing 1: **Python code of Eq. (21) for given two classes**

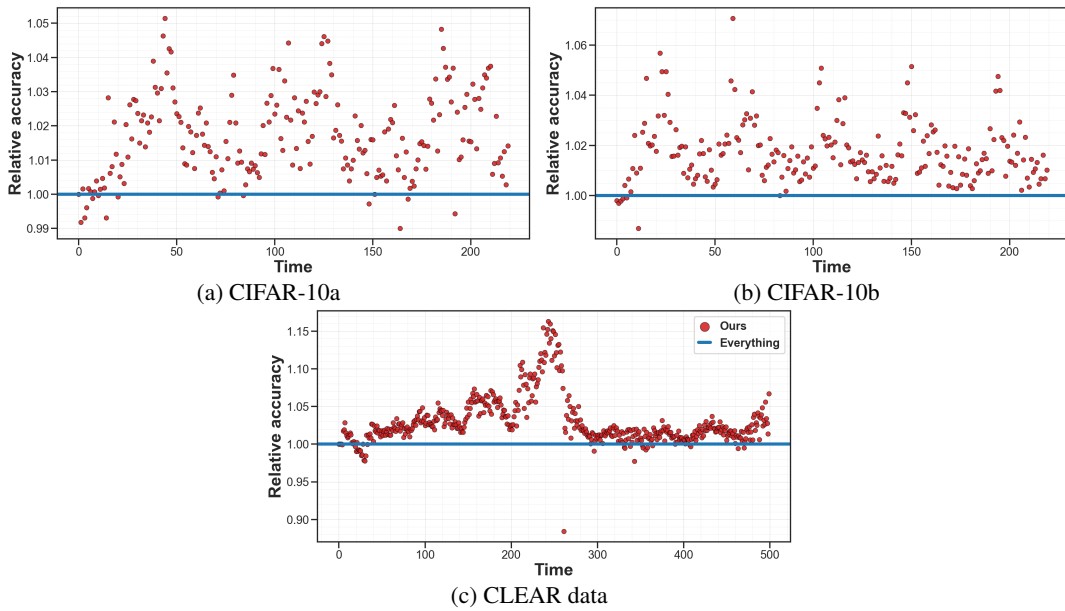

Figure 7: **Relative test accuracy achieved by our method vs Everything in continuous supervised learning benchmarks**. The experiments in these plots are exactly the same as the ones in the Fig. 1 but they are zoomed-in and we remove **Recent** and **Fine-tune** to make the plots less crowded. As mentioned before, points above the line (━━) indicate better performance than **Everything**. These plots further highlight the effectiveness of our method compared to **Everything**.

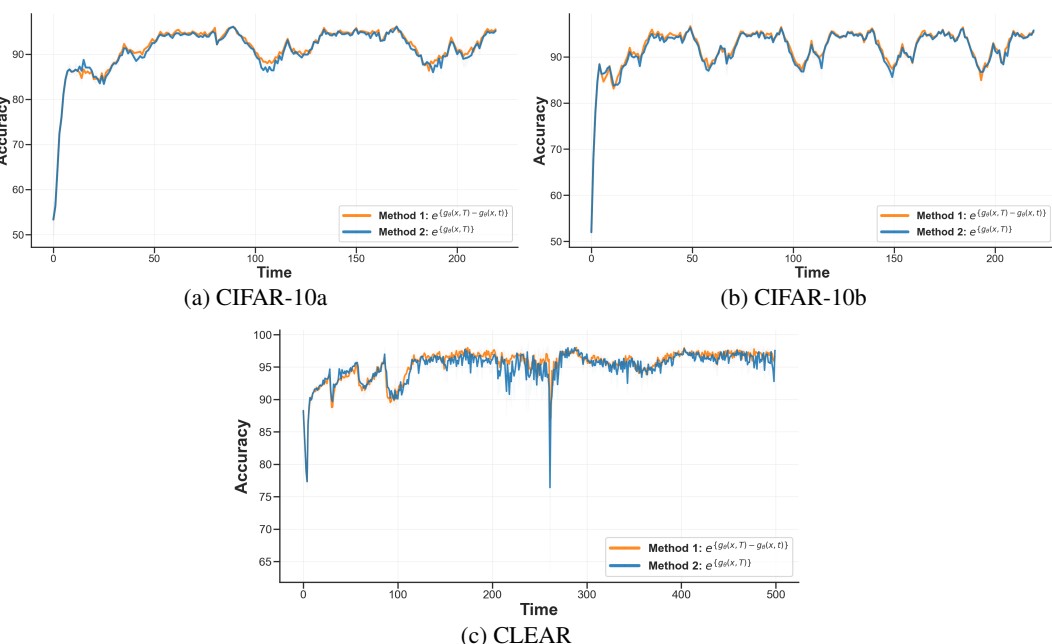

Figure 8: **Comparing our Eq. (8) (called Method 1) and Eq. (10) (called Method 2) on continuous CIFAR-10 and CLEAR benchmarks**. These results shows average test accuracy across time during test time. In each experiment, the same settings are used and the only variation is the utilization of Method 1 or Method 2 for the time-varying propensity score. These results show that both methods perform similarly, although Method 1 has a slight edge over Method 2.

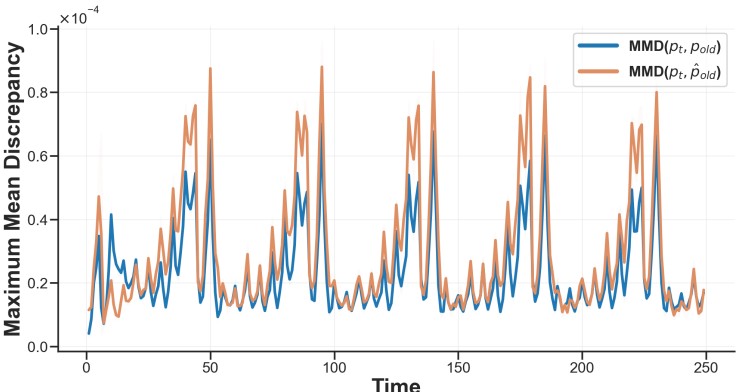

Figure 9: **Maximum Mean Discrepancy (MMD) of true distributions** $(p_t, p_{old})$ **and** $(p_t, \hat{p}_{old})$ **where** $\hat{p}_{old}$ **is constructed from past data weighted by our method**. Each (blue) point in this plot shows distance between data from $p_t(x)$ and all data from $p_{old} := \{p_{t_i(x)}\}_{t_i < t}$. Estimates for $\hat{p}_{old}$ are based on 200 images sampled at each time step from a drifting version of CIFAR-10. As the underlying distributions evolve, their MMD distance grows. To illustrate that our method produces high quality estimates, we also show distance between data from $p_t$ and all data from $\{p_{t_i}(x)\}_{t_i < t}$ which are *weighted* by our $\omega$. The small gap between blue and orange plots shows that our method produces accurate estimates.

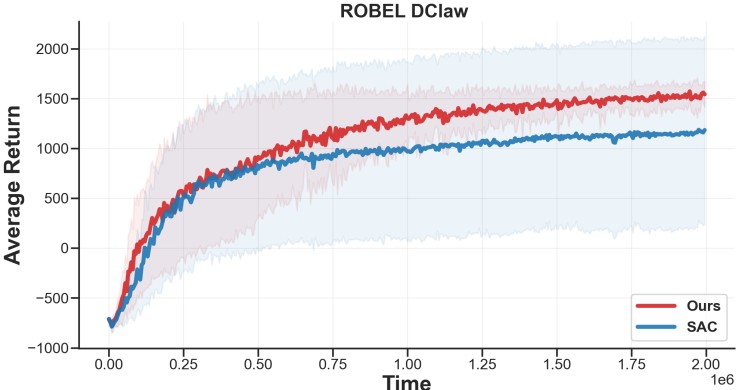

Figure 10: **Comparison of the average undiscounted return (higher is better) of our method (red) against baseline algorithm on ROBEL D'Claw**. In this environment, our method performs better than standard SAC in terms of both sample complexity and stability (compare shaded area of our method with SAC, larger means less stable). Note the only distinction between this experiment and the one in Fig. 2 is that in this experiment, we proceed to the next task after repeating each task for 25 time steps, whereas in the experiment of Fig. 2, each task is repeated only for 2 times. Our method works better in both settings and this result again verifies the effectiveness of our method in practice.

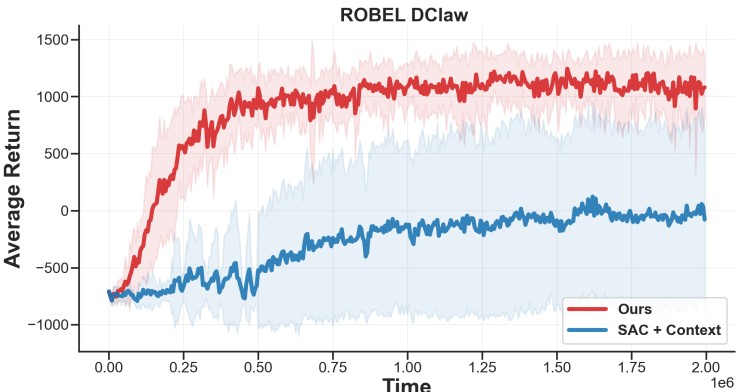

Figure 11: **Comparison of the average undiscounted return (higher is better) of our method (red) against baseline algorithm on ROBEL D'Claw**. In this environment, our method performs better than SAC + Context in terms of both sample complexity and stability (compare shaded area of our method with SAC + Context, larger means less stable). Note that the only difference between our method and baseline method is the introduction of $\omega$ term in the Q-update and all other details are *exactly* the same.

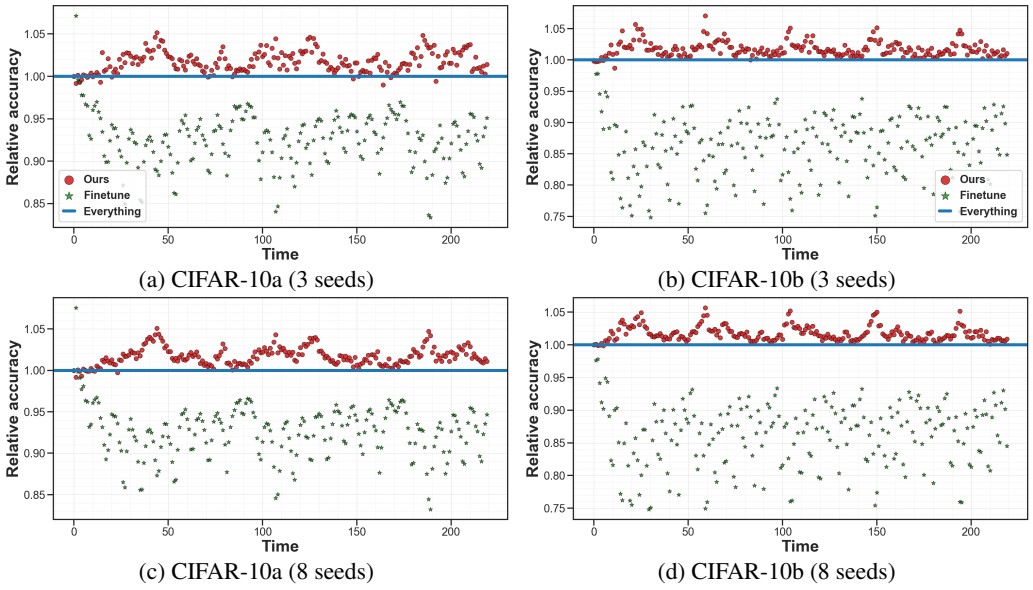

Figure 12: **Relative test accuracy achieved by our method and others vs Everything with 3 seeds against 8 seeds on continuous supervised learning benchmarks**. These results validate the consistency of our method across both 3 seeds and 8 seeds.

| Hyper-parameters for Algorithm 2 | value |
|---|---|
| Learning rate | 1e-4 |
| Number of Updates (M) | 4 |
| Batch Size | 512 |
| $g_\theta$ | 4-FC layers with *ReLU* and *BatchNorm* |
| Hyper-parameters of SAC | value |
| Random Seeds | $\{0, 1, .., 9\}$ |
| Batch Size | 256 |
| Number of $Q$ Functions | 2 |
| Number of hidden layers (Q) | 2 layers |
| Number of hidden layers ($\pi$) | 2 layers |
| Number of hidden units per layer | 256 |
| Nonlinearity | *ReLU* |
| Discount factor ($\gamma$) | 0.99 |
| Target network ($\psi'$) update rate ($\tau$) | 0.005 |
| Actor learning rate | 3e-4 |
| Critic learning rate | 3e-4 |
| Optimizer | Adam |
| Replay Buffer size | 1e6 |
| Burn-in period | 1e4 |
| $\omega$ clip | 1 |
| HalfCheetah-Wind's number of episodes to evaluate | 50 |
| Robel's number of episodes to evaluate | 5 |

Table 1: Hyper-parameters used for all reinforcement learning experiments. All results reported in our paper are averages over repeated runs initialized with *each* of the random seeds listed above and run for the listed number of episodes.

| Hyper-parameters for Algorithm 2 | value |
|---|---|
| Learning rate | 1e-4 |
| Number of Updates (M) | 200 |
| Batch Size | 512 |
| $g_\theta$ for CIFAR-10 | 5-Conv layers and 1 linear layer |
| $g_\theta$ for CLEAR | 3-FC layers with *ReLU* and *BatchNorm* |
| Hyper-parameters of CIFAR-10/CLEAR | value |
| Random Seeds | $\{0, 1, 2\}$ |
| CIFAR-10 Network | Resnet-18 |
| CLEAR Network | Linear layer |
| Samples per time step (N) | 200 |
| Learning rate | 9e-4 |
| Batch Size | 128 |
| Optimizer | Adam |
| $\omega$ clip | 1 |
| Number of training epochs per time step | CIFAR-10 (20), CLEAR (25) |

Table 2: Hyper-parameters used for all continuous supervised learning experiments for both CIFAR-10 and CLEAR tasks. All continuous supervised learning results reported in our paper are averages over three random seeds.

| Computing Infrastructure | |
|---|---|
| Machine Type | AWS EC2 - g3.16xlarge |
| CUDA Version | 11.0 |
| NVIDIA-Driver | 450.142.00 |
| GPU Family | Tesla M60 |
| CPU Family | Intel Xeon 2.30GHz |
| **Library Version** | |
| Python | 3.8.5 |
| Numpy | 1.22.0 |
| Pytorch | 1.10.0 |

Table 3: Software libraries and machines used in this paper.

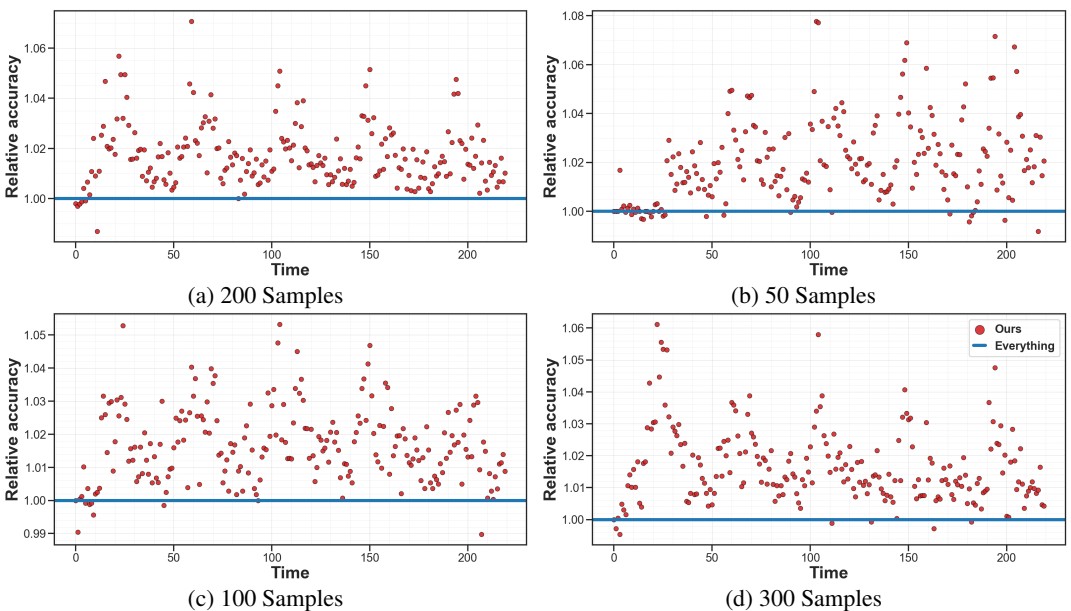

Figure 13: **Relative test accuracy achieved by our method and others vs Everything with different sample sizes on continuous supervised learning benchmarks**. Consistent results are evident across varying sample sizes, and irrespective of the size of the sample, our methods consistently outperform **Everything**.

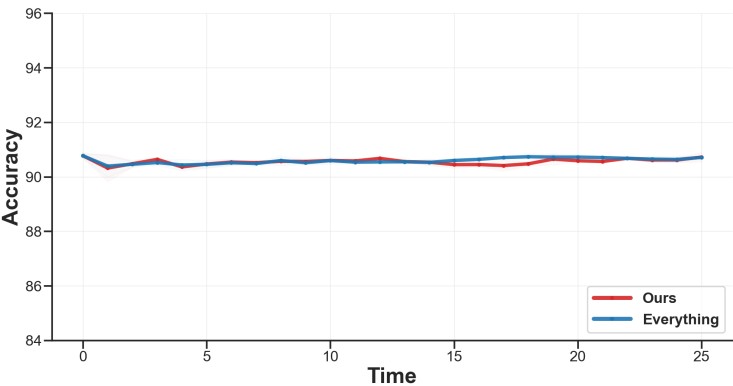

Figure 14: **Average test accuracy (higher is better) for continuous supervised learning on CIFAR-10 benchmark *without* label shift** . The x-axis indicates different test time steps and the y-axis displays the classification accuracy on test data. We see that our method and **Everything** perform similarly. This is an important observation showing our method does not negatively affect the performance when there is no shift.

