# OpenReview forum: "Time-Varying Propensity Score to Bridge the Gap between the Past and Present"
_ICLR.cc/2024/Conference — ICLR 2024 poster_

### Official Review · Reviewer_A4rR · 2023-10-31

**Soundness:** 3 good
**Presentation:** 3 good
**Contribution:** 3 good
**Rating:** 5
**Confidence:** 2

**Summary:**

The paper proposes a generalisation of propensity scores from a binary setting (i.e. one training distribution and then testing after a single discrete distribution shift) to a continuous one where data drift occurs gradually and thus the propensity scores vary over time. The method is able to detect and react to drifts without being aware of them a priori. The authors offer in-depth theoretical insights as well as an empirical analysis on both supervised as well as reinforcement learning tasks.

**Strengths:**

The proposed method can deal with continuous data drifts without knowing about them a priori by naturally extending propensity scores to continuous drift settings. Since this type of drift scenario is more likely to occur in practice than prior considered settings such as a single discrete shift, the method is likely useful in real-world problems. In the empirical evaluation, the method performs as good as or better than all considered baselines.

**Weaknesses:**

I admit I am no expert on dealing with data drift, however I am sure methods other than the proposed one and the baselines (which are all quite naive) exist. While it is good to see that on the given data the developed method outperforms the baselines, it seems to me like a rather low bar and not very surprising. I am worried that a comparison to other relevant (and more sophisticated) approaches to dealing with data drift are left unconsidered, for example I know that a huge area of work is addressing data drift issues with calibration techniques. I think in order to really show the merits of the method, the authors need to consider other state of the art methods of dealing with data drift.

**Questions:**

see weaknesses

---

> ### Author Response · Authors · 2023-11-19
> **Response to Reviewer A4rR**
>
> Thank you for your feedback and positive points about our paper. It appears to be the primary concern raised by the reviewer that there is a substantial body of work in this area and related to this problem. As we explained below, this is not accurate to the best of our knowledge and we hope the reviewer considers increasing their score after setting our response.
>
>
> > **I admit I am no expert on dealing with data drift, however I am sure methods other than the proposed one and the baselines (which are all quite naive) exist. While it is good to see that on the given data the developed method outperforms the baselines, it seems to me like a rather low bar and not very surprising.**
>
> We respectfully disagree with the reviewer. Despite reviewer claims, our baselines are indeed very strong and suitable to this problem. We want to underscore that this problem is not only highly challenging and difficult to address but also represents a novel and crucial issue, with only a limited number of effective works in this particular area ( which we already included them). We are among the first to formulate this setting and propose a simple but yet effective way for it.
>
> Nevertheless, if the reviewer can direct us to any methods applicable to this problem setting, we are happy to discuss and check whether or not they are relevant to our work. To the best of our knowledge, there is no existing method for our problem that we have not discussed.
>
> >**I know that a huge area of work is addressing data drift issues with calibration techniques. I think in order to really show the merits of the method, the authors need to consider other state of the art methods of dealing with data drift.**
>
> To the best of our knowledge, there is not such a large body of literature using calibration techniques that specifically investigate settings and problems identical to ours. More importantly, while there is a small chance that some less relevant works may not be explicitly covered in this paper, we have ensured to discuss all important, main, and critical works and include them as baselines. Finally, we are not aware of any other state-of-the-art methods known to us that go beyond the ones presented in this paper.
>
> We urge the reviewer to provide pointers to related works that discuss and study the very same problem as this paper. We are happy to discuss and include them in our paper if they deem to be relevant. Without this, it is difficult to address what the reviewer is concerned about.

---

> > ### Comment · Reviewer_A4rR · 2023-11-21
> > **Rebuttal Response**
> >
> > Dear authors,
> >
> > thank you for your response. It seems indeed that you are addressing a problem which in this identical formulation has not been well studied before. However, as I was trying to point out, and as also stated by other reviewers, there are several problems very closely related to the one you are considering, which have been studied before. In my experience, even though two problems have not the *precisely* same definition, oftentimes methods and ideas that worked well in one of them can be applied in the other with minor adjustments, and more often than not they work better than coming up with something entirely new. I'm not saying that is necessarily the case in your work, however comparing it to 1 or 2 baselines from closely related tasks would greatly strengthen your manuscript, since otherwise it is hard to tell how big of a leap forward your approach comprises.
> >
> > Since you have asked for concrete pointers, I'm going to lean out of the window here and make some suggestions, even though they might not be the most suitable ones since I am lacking a complete overview of domain adaptation works, but anyways you may get the idea:
> > - In [1], the authors propose a continual test time adaptation scenario where drift occurs gradually like in your setting. The key distinction here appears to me that the procedure is assumed to be source-free, i.e. no access to prior data is granted, just a pre-trained model. It seems to me you should be easily able to demonstrate better performance since you are able to use data from any previous timestep
> > - in the much more often considered "standard" domain adaptation scenario (e.g. [2]), access to the source data is assumed, but the drift is not gradual - you could still construct an experiment to compare these methods with yours by simply considering all prior data as the source domain and adapting to the next step
> > - a setting between the two and even closer to yours appears to me the continuous domain adaptation one that is tackled e.g. in [3], where access to the source is given and many continuously more different target domains are considered. Also here, it seems like an easy adaptation to your scenario would be possible
> > - continual learning, e.g. as considered in [4]: As other reviewers have pointed out, continual learning scenarios are closely related. Since methods dealing with these problems usually need to address catastrophic forgetting, which you do not need to consider in your case, you should be able to demonstrate having an edge over these works if applied to your scenario.
> >
> > I have found many more works in this area & I am sure you will be much better at identifying which one is most closely related and best suitable to apply to your scenario. However, I am convinced there is prior work that is easily adpated to your case, and that you should compare your method to them in order to demonstrate the merits of and the need for your approach.
> >
> >
> > [1] Wang, Q., Fink, O., Van Gool, L., & Dai, D. (2022). Continual test-time domain adaptation. In Proceedings of the IEEE/CVF Conference on Computer Vision and Pattern Recognition
> >
> > [2] Hoyer, L., Dai, D., & Van Gool, L. (2022). Daformer: Improving network architectures and training strategies for domain-adaptive semantic segmentation. In Proceedings of the IEEE/CVF Conference on Computer Vision and Pattern Recognition
> >
> > [3] Bobu, A., Tzeng, E., Hoffman, J., & Darrell, T. (2018). Adapting to continuously shifting domains. ICLR workshop track
> >
> > [4] Zenke, F., Poole, B., & Ganguli, S. (2017, July). Continual learning through synaptic intelligence. In International conference on machine learning

---

> > > ### Author Response · Authors · 2023-11-22
> > > **Re: Response to Reviewer A4rR (part 1/2)**
> > >
> > > We thank the reviewer for acknowledging the value of our work and continuing discussion with us. Having thoroughly reviewed the papers shared by the reviewer, we are confident that we haven't overlooked any significant works here as we mentioned before. Furthermore, the papers provided by the reviewer are not directly relevant to our work. Please refer to our responses below for further clarification. We hope the reviewer still considers increasing their score after seeing our latest response.
> > >
> > >
> > >
> > > >**thank you for your response. It seems indeed that you are addressing a problem which in this identical formulation has not been well studied before. However, as I was trying to point out, and as also stated by other reviewers, there are several problems very closely related to the one you are considering, which have been studied before. In my experience, even though two problems have not the precisely same definition, oftentimes methods and ideas that worked well in one of them can be applied in the other with minor adjustments, and more often than not they work better than coming up with something entirely new. I'm not saying that is necessarily the case in your work, however comparing it to 1 or 2 baselines from closely related tasks would greatly strengthen your manuscript, since otherwise it is hard to tell how big of a leap forward your approach comprises.**
> > >
> > > Thank you for your references, they are certainly interesting continual learning papers but not related to our work. We study a different problem than continual learning and there is a very clear distinction between what we do and what continual learning does. In particular, we focus on the problem of “how to select which past samples are similar to the current distribution of data”, where continual focuses on how to update models continually. We use this to predict only the immediate next task, namely $p_\{T+1\}$. Continual learning focuses on how to update a model to a new task, and it seeks to predict accurately on all past tasks. The two are totally different problems, but both are important and valuable problems. Further, we have 100s of tasks in our experiments, current continual learning methods are ill-suited to address such a large number of evolving tasks.
> > >
> > > Now, let’s discuss the papers that you have mentioned:
> > >
> > > Consider the replay buffer approach in continual learning, such as [10, 3]. This approach maintains a small buffer that stores a limited number of examples from previously encountered tasks. These stored examples are then reused during the training process for subsequent tasks. In our problem setting, it is impossible to select past data that is suitable for every time. The ideal replay buffer for each time t is totally different. So continual learning with a fixed length buffer cannot be competitive. With an unbounded buffer, continual learning cannot be better than the Everything baseline without doing some kind of propensity-based sampling.
> > >
> > > Moreover, methods like [4] and [11] are mainly focused on the issue of catastrophic forgetting which is not related to our paper.  Notably, [4] proposed a surrogate loss that regularizes the model's parameters based on the previous task's parameters. However, our approach differs significantly from [4] and [11] in that we train each model independently from scratch and we don't have access to previous task's parameters.
> > >
> > > The setup of [1] is very different from ours too as they study a different problem. In particular, they assume there is access to pre-trained weights and they do test-time adaptation, whereas our assumptions do not allow for such access and we don't have test-time adaptation. In addition, the setup of [2] is also very different from ours. [2] employs a network architecture comprising a Transformer encoder and a multi-level context-aware feature fusion decoder, implementing three training strategies and applied to segmentation problems. As it can be easily seen, not only are the settings and use cases of these methods significantly different from our problem setup, but they are also highly intricate and tailored to specific use cases.

---

> ### Author Response · Authors · 2023-11-22
> **Re: Response to Reviewer A4rR (part 2/2)**
>
> Finally, we would like to bring the reviewer’s attention to the experiments in Figure 4 of the appendix where we adapted the "Meta-Weight-net" method of [9] to our setting and ran experiments comparing it to our method. As shown in Figure 4 of the appendix, our method outperforms Meta-Weight-net significantly, which provides additional evidence of the efficacy of our proposed method.
>
> [9] Jun Shu, Qi Xie, Lixuan Yi, Qian Zhao, Sanping Zhou, Zongben Xu, Deyu Meng Meta-Weight-Net: Learning an Explicit Mapping For Sample Weighting,. NeurIPS 2019.
>
> [10] Arslan Chaudhry, Marcus Rohrbach, Mohamed Elhoseiny, Thalaiyasingam Ajanthan, Puneet K Dokania, Philip HS Torr, and Marc’Aurelio Ranzato. On tiny episodic memories in continual learning. ICML 2019.
>
> [11] Kirkpatrick, James, et. al. Overcoming catastrophic forgetting in neural networks. PNAS, pp. 201611835, March 2017. ISSN 0027-8424, 1091-6490. doi: 10.1073/pnas.1611835114.
>
> Shared by the reviewer:
>
> [1] Wang, Q., Fink, O., Van Gool, L., & Dai, D. (2022). Continual test-time domain adaptation. In Proceedings of the IEEE/CVF Conference on Computer Vision and Pattern Recognition
>
> [2] Hoyer, L., Dai, D., & Van Gool, L. (2022). Daformer: Improving network architectures and training strategies for domain-adaptive semantic segmentation. In Proceedings of the IEEE/CVF Conference on Computer Vision and Pattern Recognition
>
> [3] Bobu, A., Tzeng, E., Hoffman, J., & Darrell, T. (2018). Adapting to continuously shifting domains. ICLR workshop track
>
> [4] Zenke, F., Poole, B., & Ganguli, S. (2017, July). Continual learning through synaptic intelligence. In International conference on machine learning
>
> >**I have found many more works in this area & I am sure you will be much better at identifying which one is most closely related and best suitable to apply to your scenario. However, I am convinced there is prior work that is easily adpated to your case, and that you should compare your method to them in order to demonstrate the merits of and the need for your approach.**
>
>
> As previously mentioned, this problem setting is relatively fresh and we are among first who formulate it where we propose a simple yet highly effective method to address it. Importantly, we have written an elaborate paper and have discussed the relevant work fairly and exhaustively. We have used strong baseline methods to evaluate our work clearly, e.g., using the Everything baseline, or using Fine-tuning and we consider both RL and surprised learning. The papers that you have suggested are on continual learning and not directly related to this problem. **It is not at all fair to us to say that “you are convinced there is prior work that is easily adapted to our problem”**.   We would appreciate a pointer to any related pointer if it is missed. However, none of the above papers is relevant as we explained.

---

### Official Review · Reviewer_evvV · 2023-10-31

**Soundness:** 3 good
**Presentation:** 3 good
**Contribution:** 3 good
**Rating:** 6
**Confidence:** 4

**Summary:**

This paper proposes a new formulation of propensity scores to handle a time-evolving data distributions. Specifically, they propose to approximate the propensity scores given the seen data, the time the data was received, and the current time. Because the propensity score is conditioned on time, the authors propose this will better weight how related past data is to the current data distributions. The paper then explores the effectiveness of the estimated propensity scores through several classification tasks and in reinforcement learning baseline.

**Strengths:**

The method is simple and easily understandable to anyone familiar with variance reduction through importance sampling which are quite common throughout machine learning and reinforcement learning. This approach is seemingly novel as far as I can tell, which shows the need to explore these methods in the continual learning setting.

**Weaknesses:**

After discussing with the authors, I believe they have adequately addressed my concerns. I am updating my score to account for their changes (and the one addition I just asked for).


------ before edits -----

There are several weaknesses in this paper’s presentation and content. Some of these are less critical than others, but the overall value of the paper’s ideas are undermined by a series of confusing notation choices and overly confusing writing practices. There are also several issues with the empirical section which need clarification or addressing in subsequent submissions.

**Section 2**

1. The first weakness comes from how this work is situated in the literature, specifically the effort to distance this work from related work in continual learning. Specifically, on page 3 in Remark 2. I would argue that your setting is exactly encapsulated by continual learning frameworks. For a framework https://arxiv.org/pdf/2012.13490.pdf is a good resource. Specifically, definition 1 is a good starting point. While these definitions take a reinforcement learning perspective, they have counterparts in supervised settings as well, and can be readily adjusted to the supervised setting.
2. I’m not sure I’m understanding the construction of your dataset D, and subsequently p(t). To me, each datum is received sequentially and therefore t_i is unique for each entry. This means p(t) is just a uniform distribution for the number of timesteps (of course this is not entirely accurate as t is assumed to be from the reals and not integers, but in effect this is what I understand). Given how you use this distribution, I’m assuming this is not the case. But it is not clear what t is here, how does t relate to how the data was generated? How is your dataset constructed? This confusion continues throughout the paper including all of section 2, and when defining objectives in section 3.
3. The three lines before remark 4 are very confusing, and don’t clarify how $g_\theta(x,t)$ is estimated. Do you mean t’ instead of s’? This comes up again after remark 4 under equation 10. It is not clear how the dataset is being generated to estimate g_\theta, and the subtle difference from equation 8&9 is completely unclear (yes different distributions of the data are used, but how). It is very possible Algorithm 2 will clarify this, but I think a clarification on what the time in the original dataset (see 1) will deal with a lot of the confusion here.

**Empirical section:**

4. You did not explain how relative accuracy is defined. I’m left to assume from the first line of figure 1’s caption that they are normalized by the performance of the everything benchmark, but how is not stated in the main paper. If this is normalized according to the accuracy of the everything baseline, I think these results might be misleading. The paper would benefit from a discussion on how well the baseline is actually performing to give more perspective on the other method’s performance. It is also unclear how the models are being tested in these experiments. Are you testing the accuracy of the model on the data generated at t+1? How is this data gathered? Is this same data then added to the dataset. There are many details missing, likely related to the confusion mentioned above.understanding
5. **Recent** and **Fine-tune** are not oracles given your problem definition. If you are testing on data generated from t+1, the oracle would be a model trained on data generated from this distribution (this model would represent the best we could hope for).
6. Hyperparameters and other experimental details:
   - How were the hyperparameters chosen for your method and for the competitors? This question applies to all experiments as I don’t see it mentioned in the paper.
   - How does the amount of computation for your method relate to the amount of computation for the baselines (for both training and inference). I notice that your method has a larger batch size than the baselines for both the reinforcement learning and supervised learning settings. Why was this chosen?
   - How does your method relate in number of updates compared to the baselines? Because of the different training regimes (i.e. the inner loop w/ M for your method), it is unclear how this compares to the epochs.
   - Are the number of parameters in your models comparable to the baseline parameters.
   - In the reinforcement learning experiment what are the shaded regions signifying? Also, these should be made darker. Currently it looks like there is significant overlap in both the experiments meaning there is not a statistically significant difference between your method and SAC.


# Edits/Suggestions
- Line right after equation 4 “TV(p_{T+dt}, p_T = “
- After section 2.2, the paper shifts towards integer-valued time. I think the previous sections could benefit from this treatment as well, focusing on the simpler case of integer time and leaving real-valued time to the appendix.

**Questions:**

1. Is the first assumption the same as saying it is lipschitz smooth/continuous?

---

> ### Author Response · Authors · 2023-11-19
> **Response to Reviewer evvV (Part 1/3)**
>
> Thank you for your detailed feedback. While we are disheartened by the low score you assigned to our paper, despite recognizing its novelty, we remain hopeful that, upon reading our responses, you will reconsider and increase your score as your primary comments revolve around minor clarification questions rather than fundamental issues. Furthermore, it seems that your lower score is linked to minor presentation issues. This is surprising, as all other reviewers are content with the quality, describing it as well-written and easy to follow. Please let us know if you have more questions.
>
>
> >**The first weakness comes from how this work is situated in the literature, specifically the effort to distance this work from related work in continual learning. Specifically, on page 3 in Remark 2. I would argue that your setting is exactly encapsulated by continual learning frameworks. For a framework https://arxiv.org/pdf/2012.13490.pdf is a good resource. Specifically, definition 1 is a good starting point. While these definitions take a reinforcement learning perspective, they have counterparts in supervised settings as well, and can be readily adjusted to the supervised setting.**
>
> We think there is a simple misunderstanding here. We don’t try to distance our work from continual learning literature but rather we study a different problem and there is a very clear distinction between what we do and continual learning. In particular, we focus on the problem of “how to select which past samples are similar to the current distribution of data”. We use this to predict only the immediate next task, namely $p_\{T+1\}$. Continual learning focuses on how to update a model to a new task, and it seeks to predict accurately on all past tasks. The two are totally different problems, but both are important and valuable problems. Further, we have 100s of tasks in our experiments, current continual learning methods are ill-suited to address such a large number of evolving tasks.
>
> >**I’m not sure I’m understanding the construction of your dataset D, and subsequently p(t). To me, each datum is received sequentially and therefore t_i is unique for each entry. This means p(t) is just a uniform distribution for the number of timesteps (of course this is not entirely accurate as t is assumed to be from the reals and not integers, but in effect this is what I understand). Given how you use this distribution, I’m assuming this is not the case. But it is not clear what t is here, how does t relate to how the data was generated? How is your dataset constructed? This confusion continues throughout the paper including all of section 2, and when defining objectives in section 3.**
>
>
> Let us repeat a bit more elaborately the construction in Section 2.1. Suppose the dataset is $D =\\{x_i, t_i\\}^N_i$. The time $t$ is a modeling choice, it can be an integer or a real number. The time $t_i$ simply refers to the time at which a datum was recorded, say the Unix time. Let us suppose it is a real number, like we have done in the paper. If each datum is received at a unique time then all the $t_i$s are different. Like we described in the paper, the distribution $p(t) = \frac{1}{N} \sum_{i=1}^N \delta_{t_i}(t)$, i.e., the empirical distribution of the times recorded in the dataset. It is important to note that some times may not have any samples; this is not an issue when we take expectations/integrals over time $t$ because the distribution $p(t)$ will have zero probability over such times. If multiple samples were recorded at some instant, there would simply be two Dirac deltas in this summation. All the theory remains the same.
>
> >**The three lines before remark 4 are very confusing, and don’t clarify how  is estimated. Do you mean t’ instead of s’?**
>
> Thanks, yes, s’ should be t’. This has been fixed. Please see the response to the following question.
>
> >**This comes up again after remark 4 under equation 10. It is not clear how the dataset is being generated to estimate g_\theta, and the subtle difference from equation 8&9 is completely unclear (yes different distributions of the data are used, but how). It is very possible Algorithm 2 will clarify this, but I think a clarification on what the time in the original dataset (see 1) will deal with a lot of the confusion here.**
>
> Let us explain algorithm 2 to clarify how data is generated.
>
> To train our propensity score, we need to create a triplet $(x_j,t',t)$ for each of the samples in the dataset. In particular, if $t_j$ denotes the true time from which $x_j$ is observed, then we want our training data to have $t' = t^j$ ( and $t$ equals to a random time) in half of the training examples with label 1 and $t'$ equals random time in the other half (with $t =t_j$) with label -1.  Once this dataset is generated, the rest is to train a binary classifier using equation 19.
>
> Please let us now if this is still unclear, happy to provide more explanation.

---

> > ### Author Response · Authors · 2023-11-19
> > **Response to Reviewer evvV (Part 2/3)**
> >
> > >**You did not explain how relative accuracy is defined. I’m left to assume from the first line of figure 1’s caption that they are normalized by the performance of the everything benchmark, but how is not stated in the main paper. If this is normalized according to the accuracy of the everything baseline, I think these results might be misleading.**
> >
> > Yes, the caption defines the relative accuracy which is a standard metric in literature to present such results. We’ll repeat the exact same definition in the main text paper as well. Our result simply states that if we compare all methods with Everything how much performance decreases or increases with respect to the Everything baseline.
> >
> > The reviewer's suggestion that the results might be misleading is perplexing to us. Could you please elaborate on this point? Does the reviewer suggest that because relative accuracy is not mentioned in the body of the paper but only in the caption?
> >
> > Considering the extensive number of different experiments presented in the paper, we believe that providing detailed metrics and findings within the respective captions offers a more engaging and comprehensive reading experience.
> >
> > >**The paper would benefit from a discussion on how well the baseline is actually performing to give more perspective on the other method’s performance.**
> >
> > We are not sure if we understand the reviewer's comment here. We believe the figures clearly show how well the different methods perform compared to each other, and provide a complete overview of their effectiveness.
> >
> > >**It is also unclear how the models are being tested in these experiments. Are you testing the accuracy of the model on the data generated at t+1?**
> >
> > Yes, we test and evaluate a trained model on unseen data (i.e. not seen during training time) at time $t+1$.  This has been explained in many places in the paper, namely remark 2, section 3.1, and the last paragraph of introduction.
> >
> >
> > >**How is this data gathered? Is this same data then added to the dataset. There are many details missing, likely related to the confusion mentioned above. understanding**
> >
> > As detailed in section 3.1, each data point in the plots for the supervised learning experiments corresponds to a distinct experiment. We train a separate model from scratch for each experiment, leading to a varying number of experiments per dataset. For instance, the Gaussian dataset involves 160 distinct models and experiments, while the CLEAR dataset encompasses 500 separate experiments and models!
> > Please also see our response above about “Data generation”.
> >
> > >**Recent and Fine-tune are not oracles given your problem definition. If you are testing on data generated from t+1, the oracle would be a model trained on data generated from this distribution (this model would represent the best we could hope for).**
> >
> > Per our problem formulation, the test data is not known during training time, hence we don’t use test data to train a model in this paper. Also, as noted in the text, Recent and Fine-tune are “like” oracles in the Gaussian experiment because test time data from $t + 1$ demonstrates a closer resemblance to recent data than to other historical data.
> >
> >
> > >**How were the hyperparameters chosen for your method and for the competitors? This question applies to all experiments as I don’t see it mentioned in the paper.**
> >
> > Our method has the exact same hyper-parameters as those of Everything. Note that Fine-tune and recent use the exact same hyper-parameters as those of Everything except that they have different learning rates (this is because the lower learning rate for fine-tuning works better) and number of training epochs per time step given these are different methods than Everything. Table 1 and Table 2 in the appendix contain this information. We conducted a minimal random hyper-parameters search for the experiments using a validation set (based on Everything) in this paper and we mostly follow standard and readily available settings for the experiments whenever it is applicable. We will update section D of appendix to provide more details about the hyper-parameters and also release the code upon publication.
> >
> > >**How does the amount of computation for your method relate to the amount of computation for the baselines (for both training and inference).**
> >
> > There is no difference in the inference time. All methods have exactly the same inference time.   The "Everything" approach trains on the entire dataset, leading to more computation compared to the "Recent" approach, which utilizes only the latest data. Consequently, an epoch in the "Everything" setting necessitates more iterations due to the larger volume of data involved. Same is true for Fine-tune vs Recent. Fine-tune and Everything have similar computation time. Our method has the same computation time as Everything with additional computation to estimate propensity score.

---

> > > ### Author Response · Authors · 2023-11-19
> > > **Response to Reviewer evvV (Part 3/3)**
> > >
> > > >**I notice that your method has a larger batch size than the baselines for both the reinforcement learning and supervised learning settings. Why was this chosen?**
> > >
> > > No, our method and baselines use the exact same batch size. For RL experiments, we use the exact same batch size as SAC which is 256 and for supervised learning we use 128 as shown in Table 1 and Table 2. The batch sizes for fitting the propensity score is 512 in both supervised and reinforcement learning. We would like to note that **all** training details are exactly the same between our method and and the base method.
> > >
> > >
> > > >**How does your method relate in number of updates compared to the baselines? Because of the different training regimes (i.e. the inner loop w/ M for your method), it is unclear how this compares to the epochs.**
> > >
> > > To demonstrate that the observed performance improvements are attributable to the proposed method itself and not to any unfair advantages in hyperparameter settings or other factors, we employed the same hyper-parameters (e.g. number of updates, etc.) and training data for both our method and the baseline methods. The results of our method could have potentially yielded even greater improvements had we engaged in specific hyperparameter tuning for our method, a step we intentionally omitted.
> > >
> > > >**In the reinforcement learning experiment what are the shaded regions signifying? Also, these should be made darker.**
> > >
> > >
> > > Following RL's literature for continuous control, we used 10 different random seeds to average each RL experiment, and the shaded areas represent the standard deviation across 10 random seeds. While some instances exhibit overlapping shaded areas, our method consistently demonstrates lower variation across runs, indicating its better stability. This is particularly crucial for methods susceptible to instability due to distributional shifts between the policy used for data collection and the policy undergoing optimization (a.k.a. off-policy methods in RL). Our results effectively demonstrate the ability of our method to lessen the detrimental effects of off-policyness.
> > >
> > >
> > > >**Is the first assumption the same as saying it is lipschitz smooth/continuous?**
> > >
> > >
> > >
> > > Not really, we use Markov processes to highlight a crisp mathematical way of thinking of drifting data distributions. The main point we make there is that if the generator of the stochastic process A has eigenvalues whose magnitude is small, then the data evolves slowly. If the generator has large positive eigenvalues, then the stochastic process quickly reaches a steady state and data does not drift anymore. Only in the case when A has large negative eigenvalues, the data changes both quickly and does not reach a steady-state. The setting of the Markov process (all finite-memory stochastic processes can be written as one) therefore allows us to explicitly understand the kind of problems for which the drift estimator is well-suited. In principle, it is possible to take a stream of data and check the eigenvalues of A to see if the problem satisfies the setting of this paper—this can be of immense practical value.
> > >
> > >
> > > Please let us know if you have more questions.

---

> ### Comment · Reviewer_evvV · 2023-11-21
> **Response**
>
> Thank you for the detailed response! I did not mean to dishearten you from the low score, but didn’t think the paper as presented was ready for publication and wanted to give a clear signal. It is surprising that my review is quite different from others, which might mean I was misreading some fundamental part of the paper which threw me off.
>
> I’m always looking for clarity in a paper so that I can exactly replicate the results without many assumptions, and the many questions left by your paper (as indicated in my review) left me with little confidence to be able to do this. I know some of these comments come off as me being pedantic, but it is to try and make a paper clearer not as a means to dishearten or discourage.  Thank you for your clarifications, as they do help.
>
> I still find myself grasping to understand how g_theta is being estimated. I think the key piece of information that I was missing was that t and t’ are randomly sampled from some distribution? Where do these new values come from (because they are clearly not just t_j)? If this is clarified it would make the rest of my questions self-answerable.
>
> Some other comments:
>
> **No access to t+1, always testing against t+1**
> These comments are coming from a place of confusion with your method. I know you are proposing to estimate distributions into the future. I was trying to clarify whether I was misunderstanding your motivation, not understanding the method, or missing something else. Thank you for the clarification.
>
> **Relative Accuracy:**
> 1. I appreciate this definition being added to the paper. While it is standard in the literature, these details still merit appearing in the text (even if relegated to the appendix). This comes from a desire to reduce the barrier of entry of the field.
> 2. I mentioned this metric being misleading not in an effort to suggest you were lying, just as a desire to understand the absolute performance of our models. While we can make a comparison across methods using the relative measure, this only gives us an ordering of algorithms. If all the algorithms perform poorly, then the relative performance quickly becomes a meaningless metric.
>
> **Baselines:** I suggest the oracle baseline not as a true competitor but as a way to understand “what is the best our methods could ever hope to do”. This  again is me wanting to understand more about the absolute performance.
>
> **Shaded Regions:** This can vary quite a bit from paper to paper. Sometimes shaded regions are standard error, sometimes they are 95% confidence intervals calculated w/ bootstrap. I did not see this specified in this particular paper, and again comes from me asking the question “could I recreate these results”.
>
> **Misunderstanding of the hyperparameters** Thank you for the clarifications! I had read these tables as changes only to your method, and not all the methods,
>
> After reading my original review and looking at the paper again, there were obvious misunderstandings I had but I do believe there is more this paper could have done for clarity. I agree with the authors that my original score was a bit tough.  I think if the authors could clarify where t and t’ come from, and address my additional comments I will be happy with the state of the paper.

---

> > ### Author Response · Authors · 2023-11-22
> > **RE: Response to Reviewer evvV**
> >
> > We are glad that our responses clarified some of the reviewer's questions and helped to highlight the contribution of our paper. Moreover, we certainly thank the reviewer for engaging in open-minded discussion with us and helping us to improve our paper further. Our objective has been to provide comprehensive details to ensure the paper is a compelling read, particularly given the fresh problem setting, which can be very valuable for the community. It is rare to see a paper to cover both RL and supervised settings in such a thorough way to show the efficacy of a proposed method. Although we were surprised by the reviewer's initial very low score, we acknowledge that there is still space for further refinement in the clarity aspect of the paper. *However, it's important to note that these improvements are relatively minor (considering reviewer’s comments) and can be easily addressed*.
> >
> > We still remain hopeful that you will increase your score after these exchanges.
> >
> >
> > >**Relative Accuracy and baselines**
> >
> > Thank you for your clarification.  We examined many different methods for visualizing the results of our experiments. Demonstrating the results of our experiments poses a unique challenge since they don't consist of a single number that can be displayed in a single plot. Instead, we have to represent hundreds of results in a single plot for each benchmark (e.g. Gaussian experiment in Figure 1 is 160 different experiments, CLEAR is about 500 different experiments, etc. ). That being said, to show that our baselines are actually strong and they don’t have poor performance and answer the reviewer's concern, we have added new  visualizations of Figure 1 to the appendix where we utilize classification accuracy in y-axis instead. Please see Figure 15 in page 23 of the updated paper. We’ve added these plots to further clarify the significance of our results.
> >
> >
> > >**Shaded Regions:**
> >
> > That is a fair point, thanks. We follow RL literature where the shaded area is standard deviation. We’ll update the text to make sure that this is clear.
> >
> > >**I think the key piece of information that I was missing was that  t and t’ are randomly sampled from some distribution? Where do these new values come from (because they are clearly not just t_j)? If this is clarified it would make the rest of my questions self-answerable.**
> >
> >
> > Let us use Algorithm 1 to answer this question. For every data point $x_j$, we know when it is recorded $t_j$; this is quite reasonable in any real problem, e.g., when a sensor records the room temperature. Now, to estimate the propensity score, we need to construct a triplet $(x_j,t',t)$ for each datum in the dataset. If $t_j$ denotes the true time at which $x_j$ was observed, then we want the dataset of our propensity estimator to consist both the correct time of $x_j$, i.e., $t_j$ and incorrect time of $x_j$, e.g., a random time chosen from among all recorded times. We therefore create a dataset with 50% samples labeled +1 as (x_j, t’=t_j, t) and another 50% samples labeled -1 as (x_j, t’, t=t_j).
> >
> > We hope this clears things out, but please let us know if there are more questions.

---

> > > ### Comment · Reviewer_evvV · 2023-11-22
> > > **Response**
> > >
> > > Excellent!
> > >
> > > You have cleared everything up I believe. One final addition I would implore the authors to make is to add the statement that t and t' are sampled from the set of times from the original dataset. That included in the main text would have prevented the confusion I faced.
> > >
> > > I also really appreciate the additions to the text to add to the clarity, and make sure all experimental details are present.
> > >
> > > I'll update my score.

---

> ### Author Response · Authors · 2023-11-23
> **Thank you for increasing your score**
>
> Thank you for revising and increasing your score and acknowledging the contribution and value of our paper.
> We are grateful for the reviewer's scholarly interaction with us which we used to further improve our paper.
>
> Sure, we'll update the final version with clarification about t and t'.
> Please let us know if you have more questions. Thank you.

---

### Official Review · Reviewer_bXrx · 2023-11-01

**Soundness:** 4 excellent
**Presentation:** 3 good
**Contribution:** 3 good
**Rating:** 8
**Confidence:** 2

**Summary:**

This paper proposes to learn a time-vary propensity score to combat distribution shift in learning problems in which the distribution of data evolves over time. At a high-level, this method learns a weighting function $\omega$ which places greater weight on historic data that evolved similarly in the past. Empirically, the time-varying propensity score increases classification accuracy in supervised learning problems and outperforms other propensity score baselines. Moreover, it improves the data efficiency in RL tasks.

**Strengths:**

1. The paper is well-motivated and well-written. The problem, solution, and novelty, are all clearly discussed in the intro.
1. The proposed method is sufficiently general to be applied to both supervised and reinforcement learning problems.
1. The experimental setup is very clear and provides intuition or how different baselines should perform in different settings.

**Weaknesses:**

1. The baselines considered for the supervised learning experiments are reasonable, but the RL experiments don't consider these baselines. As written, it's unclear if the time-vary propensity score is any better than simpler methods. Also, ROBEL DClaw is missing the SAC + Context baseline.

1. At the end of Section 2 "Approach," it would aid the reader to include a brief summary of the line of thinking outlined in sections 2.1-2.3. I found Section 2 far easier to understand on my second read largely because I already had an idea of the arguments that followed.

1. It would be useful to give a brief summary of the ablation findings (appendix C) in the main paper. For instance, mention that the proposed method performs just as well as Everything when there is no shift,

1. The shaded regions for the RL curves should be darker; as of now, they are very difficult to see.

**Typos:**
1. Section 2.1: "for some time large final time T"
2. Missing right parenthesis after Eq. 4: TV(p_{T+dt}, p_T)
3. Section 2.2 last paragraph: "is subtle when deals with data"
4. Just before Remark 5: "We call it “Finetune” make the objective"
5. Section 3.1.3 last paragraph: "We run this experiments"

**Questions:**

1. Could the authors explain the CIFAR shifts described by Eq. 21? I'm hoping for a plain English description of the shift.

1. Could the authors comment on the computation expense of each method? I imagine they're all similar in computation cost, since their losses have similar forms.

---

> ### Author Response · Authors · 2023-11-19
> **Response to Reviewer bXrx (Part 1/2)**
>
> Thank you for your positive, detailed and valuable feedback. We hope you'll consider increasing your score after reading our responses as we believe we have addressed all your comments. Please let us know if you have more comments.
>
>
> >**The baselines considered for the supervised learning experiments are reasonable, but the RL experiments don't consider these baselines. As written, it's unclear if the time-vary propensity score is any better than simpler methods.**
>
> Thanks for the question. The distinct nature of RL and supervised learning methods necessitates modifications to the underlying RL algorithm to accommodate the same baselines as supervised learning. In particular, SAC, an off-policy algorithm, utilizes a replay buffer to store data during exploration and samples uniformly during training. Implementing a baseline like fine-tune in SAC necessitates an additional replay buffer that only retains recent data, introducing further alterations and new hyperparameters to the SAC algorithm. In our RL experiment, we design experiments such that it requires minimal changes to the original RL method and it doesn't change how the underlying RL algorithm works as our goal in this paper is to show rather applicability and efficacy of our method in RL, rather than proposing a new RL algorithm.
>
> >**Also, ROBEL DClaw is missing the SAC + Context baseline.**
>
> No, it is not missing. We already included this in the Figure 11 of appendix.  We also have another additional experiment for ROBEL DCLAW where tasks are sampled differently to show the effectiveness of our method in another setting. Figure 10 shows the result of this experiment. This experiment is another piece of evidence that shows the effectiveness of our method.
>
> We also like to note that when we combine the SAC or SAC + Context with our time varying propensity score, we see that our method consistently demonstrates lower variation across runs, indicating its better stability ( see Figure 10 and 11 for examples). This is particularly crucial for RL methods susceptible to instability due to distributional shifts between the policy used for data collection and the policy undergoing optimization (a.k.a. off-policy methods in RL). Our results effectively demonstrate the ability of our method to lessen the detrimental effects of off-policyness of off-policy methods like SAC.
>
> >**It would be useful to give a brief summary of the ablation findings (appendix C) in the main paper. For instance, mention that the proposed method performs just as well as Everything when there is no shift,**
>
> That is an excellent suggestion and it is an easy fix. Due to page limits and the extensive number of experiments conducted in this paper, only a portion of them could be presented in the main body and the additional experiments are relegated to the appendix. We will ensure that the final version of the paper text will be updated to direct readers to these additional experiments and we also provide more information about these experiments in the main text. We will update the paper based on this suggestion.
>
> >**The shaded regions for the RL curves should be darker; as of now, they are very difficult to see.**
>
> That is a very easy fix and we've now updated the figures.
>
> >**Could the authors explain the CIFAR shifts described by Eq. 21? I'm hoping for a plain English description of the shift.**
>
> This formula describes the label shift used in our experiments. At each time t, the task consists of images from two particular classes, say class 6 (10% samples) and class 7 (90% samples). In this case, at the step (time t+1) causes a new class to be added to the task, i.e., it consists of class 7 (90%) and class 8 ($10%$). At time t+2, the task consists of class 7 (80% samples) and class 8 (20% samples). At any time t, the test data consists of images sampled from the task at time t+1. It is important to note that there is a clear distribution [label] shift between the train and test distributions due to this construction and therefore it is an extremely direct test of our proposed approach. This shift is also gradual, in some cases the label distribution changes and in some changes a new class is introduced, but the new class only accounts for 10% samples at the instant of introduction. In simple words, the data distribution consists of two classes and the support of this distribution shifts over all the 10 CIFAR-10 classes, as it gradually undergoes label shifts.
>
> Also, there is a typo in that formula where H should be T.  This has been fixed.

---

> > ### Author Response · Authors · 2023-11-19
> > **Response to Reviewer bXrx (Part 2/2)**
> >
> > >**Could the authors comment on the computation expense of each method? I imagine they're all similar in computation cost, since their losses have similar forms.**
> >
> > The "Everything" approach updates on the entire dataset, leading to more computation compared to the "Recent" approach, which utilizes only the latest data. Consequently, an epoch in the "Everything" setting necessitates more iterations due to the larger volume of data involved. Same is true for “Fine-tune” vs “Recent”. “Fine-tune” and “Everything” have similar computation time. Our method has the same computation time as “Everything” with additional computation to estimate propensity score.
> >
> >
> > >**Typos.**
> >
> > Thank you. We have fixed these issues in the paper
> >
> > Please let us know if you have more questions.

---

> > > ### Comment · Reviewer_bXrx · 2023-11-22
> > >
> > > I appreciate the authors' response! I now understand the choice of baselines for the RL experiments -- and thank you for pointing out the additional RL experiments in the appendix. I feel that all of my comments have been addressed.
> > >
> > > I want to note that I'm not well-versed enough in the related work to provide a thorough assessment of the work's novelty and certain aspects of the empirical setup (related to reviewer LkU5 and A4rR's comments), though I do feel okay raising my score. Although the topic of this submission is a bit outside my expertise, I did find the paper easy to follow, the topic well motivated, and the proposed method reasonable.

---

> > > > ### Author Response · Authors · 2023-11-23
> > > > **Thanks for increasing your score**
> > > >
> > > > Thank you for revising and increasing your score. We are glad to see that the reviewer sees the value and contributions of our paper. Please let us know if you have more questions.

---

### Official Review · Reviewer_LkU5 · 2023-11-06

**Soundness:** 4 excellent
**Presentation:** 3 good
**Contribution:** 3 good
**Rating:** 6
**Confidence:** 4

**Summary:**

The authors tackle the problem of gradually evolving data which is an important problem in many real world settings. The paper introduces a time-varying propensity score that can detect gradual shifts in the distribution of the data. This helps with prioritizing samples from past data that evolved in a similar fashion. The proposed approach considers optimizing a beta-weighted version of the training loss. Previous research has mainly focused on handling single shifts between training and test data or limited forms of shift involving changes in features, labels, or the underlying relationship between them.

One key advantage of this method is its ability to automatically detect data shifts without prior knowledge of their occurrence.The proposed algorithm seems versatile in that it  can be applied in different problem domains, including supervised learning and reinforcement learning, and can be integrated with existing approaches. The authors conducted extensive experiments in continuous supervised learning and reinforcement learning to demonstrate the effectiveness of their method. The broader impact of this research is in enhancing the robustness of machine learning techniques in real-world scenarios that involve continuous distribution shifts.

**Strengths:**

Typically the propensity score considered doesn’t account for the time varying aspect - the work offers a sound approach to do so. The method exploits the initial unconstrained distribution of data as a  “gauge” to choose $p_0(x)$ differently. With this, they build a model $p_t(x)$ as deviations from the marginal on $x$.

*   The proposed method considers a learning theoretic perspective to minimize the risk on the immediate next task after T tasks until that point.  The work studies a simple situation where we model the time-varying propensity score using an exponential family.
*    The experiments cover a broad empirical comparison in both continuous supervised and reinforcement learning including synthetic data sets with time-varying shifts, continuous supervised image classification tasks utilizing CIFAR-10 and CLEAR datasets, and a simulated robotic environment alongside MuJoCo.
*    The empirical evaluation shows  1) Relative test accuracy achieved by the proposed  method is consistently better than other methods in continuous supervised learning benchmarks. 2) The performance in RL is particularly noteworthy wherein the proposed method can effectively model how data evolve over time with respect to the current policy without requiring to have access to the behavior policies that were used to collect the data. This is achieved by a simple introduction of a weighting scheme as proposed in existing loss.

**Weaknesses:**

Two major weaknesses/limitations of this work are:

*    The assumption that data evolves gradually which might not be true in many real world situations and limits the scope of application of the work. Consider for e.g. the sudden shifts in the data distribution. Would the proposed approach be extendable to such settings? What do we need to cover this setting as well?
*   Evaluating on test data seems very counterintuitive as typical settings would not have this privileged information except during training. Can you provide how this specific choice of evaluation can be overcome? I find it quite limiting to do so.

**Questions:**

Please see some of the questions in weaknesses.  A few other questions/clarifications are:

How is the `Finetune` baseline different from typical online-meta learning approaches? Finetune  typically would continue learning  a hypothesis on all past tasks and then adapts it further using data from the new task at hand that is in round T $p_T$. for e.g. in [1] and [2].
 Is it primarily that in your method the desiderata is essentially different i.e  to obtain a small risk on pT +dt — as opposed to obtaining a small population risk on all tasks. Is it possible to get the best of both worlds instead i.e. get small risk on the latest task/future task while maintaining low risk across ALL tasks that the agent has seen, as otherwise there is a recency bias to only caring about the immediate next data point.

When you claim that “, existing methods for these settings do not employ time-varying propensity weights like we propose here.” Can you comment how does your work position with respect to online meta learning settings such as [1] and [2] where the methods are also able to “automatically detect data shifts without prior knowledge of their occurrence” without the knowledge of the number of tasks etc.


[1] Nguyen, Nicolas, and Claire Vernade. "Lifelong Best-Arm Identification with Misspecified Priors." Sixteenth European Workshop on Reinforcement Learning. 2023.

[2]  Khetarpal, Khimya, et al. "POMRL: No-Regret Learning-to-Plan with Increasing Horizons." arXiv preprint arXiv:2212.14530 (2022).

---

> ### Author Response · Authors · 2023-11-19
> **Response to Reviewer LkU5 (Part 1/2)**
>
> Thank you for your positive, detailed, and valuable feedback. We hope you'll consider increasing your score after reading our responses as we believe we have addressed all your comments. Please let us know if you have more comments.
>
>
> >**The assumption that data evolves gradually which might not be true in many real world situations and limits the scope of application of the work.**
>
>
> While we agree with the reviewer that there are other kinds of shifts that can change data, the gradual shift assumption is relevant for many real-world scenarios where data is collected from evolving distributions that change over an extended period. For instance, drift in environmental variables such as temperature caused by day/night or seasonal changes affects both the forecasting systems and the systems that collect the data [1]. There are also many examples in oceanography, e.g., currents and other relevant oceanographic variables such as salinity and temperature in a coastal area change slowly with time [2,3] and modifying the forecasts for these variables is challenging because of the drift in the data. Even for problems in healthcare, there are many situations where data drifts gradually, e.g., the different parts of the brain atrophy slowly as the brain ages for both healthy subjects and those with dementia; tracking these changes and distinguishing between them is very useful for staging the disease and deciding treatments [4,5]. Viruses can also mutate and drift over time, which can make them resistant to existing treatments [6,7,8]. We hope that this addresses your concern about the motivation of the problem: gradual drift data is indeed a problem with broad applications. There is enormous economical and intellectual value in addressing graduate data drift.
>
> [1] Learning under Concept Drift: A Review. IEEE Transactions on Knowledge and Data Engineering, vol. 31, no. 12, pp. 2346-2363, 1 Dec. 2019, doi: 10.1109/TKDE.2018.2876857.
>
> [2] Operational Forecast System, Center for Operational Oceanographic Products and Services. https://tidesandcurrents.noaa.gov/models.html
>
> [3] Changing sea levels: effects of tides, weather and climate. David Pugh. Cambridge University Press.
>
> [4] Saito Y, Kamagata K, et al., Temporal Progression Patterns of Brain Atrophy in Corticobasal Syndrome and Progressive Supranuclear Palsy Revealed by Subtype and Stage Inference (SuStaIn). Front Neurol. 2022.
>
> [5] Rongguang Wang, et al.,  Adapting Machine Learning Diagnostic Models to New Populations Using a Small Amount of Data: Results from Clinical Neuroscience. 2023. https://arxiv.org/abs/2308.03175
>
> [6] Ewen Callaway. The coronavirus is mutating — does it matter?, September 2020.
>
> [7] William T Harvey, et al., COVID-19 Genomics UK (COG-UK) Consortium, et al. Sars-cov-2 variants, spike mutations and immune escape. Nature Reviews Microbiology, 19(7):409–424, 2021.
>
> [8] C.J. Russell. Orthomyxoviruses: Structure of antigens. In Reference Module in Biomedical Sciences. Elsevier, 2016. ISBN 978-0-12-801238-3.
>
>
> >**Consider for e.g. the sudden shifts in the data distribution. Would the proposed approach be extendable to such settings? What do we need to cover this setting as well?**
>
>
> This is a good question. Yes, our method can be potentially extended to such a setting as long as there is data in the past which are similar or close to the current sudden shift. If the shift is sudden, e.g., if there is no predictable pattern in how the data evolved suddenly and the new data is very different from all past data, our method and in fact no other method will work without explicitly using test data to adapt.
>
>
> >**How is the Finetune baseline different from typical online-meta learning approaches? Finetune typically would continue learning a hypothesis on all past tasks and then adapts it further using data from the new task at hand that is in round T . for e.g. in [1] and [2].**
>
> That is a great question. The main difference between Finetune and online meta learning is that online meta learning methods are formulated as meta-learning problems and they use test data for the adaptation. For the fine-tune baseline, we don’t have access to the test data to do adaptation and we don't formulate our problem like a meta-learning objective.
>
> >**Is it primarily that in your method the desiderata is essentially different i.e to obtain a small risk on pT +dt — as opposed to obtaining a small population risk on all tasks.**
>
> Exactly, that is correct. Afterall, that is the most realistic problem in situations where the data distribution changes; the model must work well for the new data, not on average over all past data.

---

> ### Author Response · Authors · 2023-11-19
> **Response to Reviewer LkU5 (Part 2/2)**
>
> >**Evaluating on test data seems very counterintuitive as typical settings would not have this privileged information except during training. Can you provide how this specific choice of evaluation can be overcome? I find it quite limiting to do so.**
>
>
> There seems to be a simple misunderstanding here. We are not sure why the reviewer finds the test setting to be counterintuitive and why they think our method has access to privileged information during the test time which is otherwise only available during training time. Could you please provide more details on these matters so that we can address their concerns effectively?
>
>
> In the meantime, let us clarify further the test and training time settings. For the purposes of calculating the propensity score, we create the datasets $\\{(x,+1), (x’,-1)\\}$ using the training samples. This estimator is run on all the training samples from $p_T$ and $p_t$ for $t < T$ to calculate the propensity of each of the $p_t$ with respect to $p_T$. **No test samples are ever used during training time (as is appropriate)**. A model is trained using the weighted version of the dataset (which now has samples from t = 0 to t = T) and is used to make inference on samples from $p_{T+1}$. No samples from $p_{T+1}$ are used to train the propensity estimator or the model.
>
> Moreover, we emphasize that there is no information about when, where, or what shifts happen in the past, present, or future during both training and test times and our method doesn’t know them as prior knowledge. More importantly, this is why it is remarkable that our method can identify continuously changing data distributions without knowing when or how the distribution changed.
>
>
>
> >**Is it possible to get the best of both worlds instead i.e. get small risk on the latest task/future task while maintaining low risk across ALL tasks that the agent has seen, as otherwise there is a recency bias to only caring about the immediate next data point.**
>
>
> That is certainly possible and that is a good suggestion. In this work, we mainly focus on the setting where the problem is not formulated as a meta-learning objective and there is no access to the test-data to do adaptation which is closer to real-world setting. Our results confirm the effectiveness and applicability of our proposed method. However, combining our method with meta-learning based techniques can be a future research direction, but that is beyond the scope of this paper. In general, one should expect we cannot both have a small risk on the latest task and small risk on all past tasks, a number of papers have argued this, e.g., [10]. There can be situations when the tasks have solutions that are very similar to each other, and in such cases one will indeed have small risk on the latest task as well as past tasks (but there is also no effective distribution shift in such problems).
>
> Finally, we would like to bring the reviewer’s attention to the experiments in Figure 4 of the appendix where we adapted the "Meta-Weight-net" method of [9] to our setting and ran experiments comparing it to our method. As shown in Figure 4 of the appendix, our method outperforms Meta-Weight-net significantly, which provides additional evidence of the efficacy of our proposed method.
>
> [9]  Jun Shu, Qi Xie, Lixuan Yi, Qian Zhao, Sanping Zhou, Zongben Xu, Deyu Meng Meta-Weight-Net: Learning an Explicit Mapping For Sample Weighting,. NeurIPS 2019.
>
>
> [10] Rahul Ramesh, Pratik Chaudhari, Model Zoo: A Growing "Brain" That Learns Continually,. ICLR 2022.
>
>
>
> Please let us know if you have more questions.

---

### Author Response · Authors · 2023-11-19
**General Response to all reviewers**

We thank all of the reviewers for their valuable feedback and we are glad to receive it, and certainly glad to see that reviewers believe our paper to be well-written, well-motivated, novel, simple but effective, the experiments thorough and comprehensive, and its potential application in real-world problems.


To summarize our contributions:

- Distribution shift is a major theme in the recent literature on machine learning; it has received wide attention, both theoretical and empirical. We have argued in this paper that one major class of distribution shift problems, namely where data distributions shift gradually across time, can be addressed effectively by modeling how the distribution shifts across time. The technical novelty here is to think of data being drawn from a non-stationary stochastic process and setting up a problem where the learner only uses all data up to some time T to make predictions on data infinitesimally from the future, from time T+dt. This is a new setting and has never been investigated before in such a comprehensive way and like our paper.
- We proposed a simple yet effective time-varying importance weight estimator to account for gradual shifts in the data distribution. This estimator can be used to make predictions on future data without any explicit knowledge of when and how the distribution changes across time.

- We have developed a general approach that can be used very effectively in many different settings---from supervised learning to reinforcement learning---and it can also be used in tandem with many existing approaches for these problems such as online learning and meta-learning.

We also made minor updates to the paper to address some of the reviewers’ comments and submitted the revised version. We have addressed each reviewer's concerns in the detailed responses provided below.

---

### Meta-Review · Area_Chair_U5Ck · 2023-12-05

**Metareview:**

This paper looks at the problem of gradually evolving data, asking the question of how to identify samples from the past that are similar to those coming from the current (evolved) distribution of data. Such an approach is extensively evaluated in both supervised and reinforcement learning problems. I am recommending the paper to be accepted as it tackles an important problem that in a sense has not been very studied by the community. I welcome this new perspective the paper provides. The authors did a very good job addressing the concerns raised by the reviewers during the discussion phase.

**Justification For Why Not Higher Score:**

This paper seems to be one of those papers that are introducing a new framework and because of that they receive some pushback because it is different from what people are used to. In that sense, there's a real anchoring effect given that it received a 5, and 2 6s (besides the 8, of course). I think the paper should be accepted but I'm not confident in giving it a higher score. Many questions were raised about presentation, empirical practices, and so on. It seems they have mostly been addressed, but it was such a big swing from the original assessment that I think making it a poster is the right decision.

**Justification For Why Not Lower Score:**

This paper seems to be one of those papers that are introducing a new framework and because of that they receive some pushback because it is different from what people are used to. The only reviewer who thinks the paper should be rejected didn't make a good case and in a sense they are not even confident in their score.

---

### Decision · Program_Chairs · 2024-01-16

Accept (poster)